

# Imaging groundwater infiltration dynamics in karst vadose zone with long-term ERT monitoring

Arnaud Watlet[1,2], Olivier Kaufmann[1], Antoine Triantafyllou[1,3], Amaël Poulain[4], Jonathan E. Chambers[5], Philip I. Meldrum[5], Paul B. Wilkinson[5], Vincent Hallet[4], Yves Quinif[1], Michel Van Ruymbeke[2], Michel Van Camp[2]

[1]Geology and Applied Geology Unit, Faculty of Engineering, University of Mons, Place du Parc 20, 7000 Mons, Belgium.
[2]Seismology-Gravimetry, Royal Observatory of Belgium, Avenue Circulaire 3, 1180 Uccle, Belgium.
[3]Laboratoire de Planétologie et Géodynamique — Nantes (LPGN), UFR Sciences et Techniques, Université de Nantes, UMR-CNRS 6112, Rue de la Houssinière 2, BP92208, 44322 Nantes Cedex 3, France
[4]Department of Geology, University of Namur, Rue de Bruxelles 61, 5000 Namur Belgium
[5]Geophysical Tomography Team, British Geological Survey, Nottingham NG12 5GG, UK

*Correspondence to*: Arnaud Watlet (arnaud.watlet@umons.ac.be)

**Abstract.** Water infiltration and recharge processes in karst systems are complex and difficult to measure with conventional hydrological methods. Especially, temporarily saturated groundwater reservoirs hosted in the vadose zone can play a buffering role in water infiltration. This results from the pronounced porosity and permeability contrasts created by local karstification processes of carbonated rocks. Analyses of time-lapse 2-D geoelectrical imaging over a period of three years at the Rochefort Cave Laboratory (RCL) site in South Belgium highlight variable hydrodynamics in a karst vadose zone. These data were compared to conventional hydrological measurements (drip discharge monitoring, soil moisture and water conductivity data sets) and a detailed structural analysis of the local geological structures providing a thorough understanding of the groundwater infiltration. Seasonal changes affect all the imaged areas leading to increases in resistivity in spring/summer attributed to enhanced evapotranspiration, whereas winter is characterised by a general decrease in resistivity associated with a groundwater recharge of the vadose zone. Three types of hydrological dynamics, corresponding to areas with distinct lithological and structural features, could be identified via changes in resistivity: (i) upper conductive layers, associated with clay-rich soil and epikarst, showing the highest variability related to weather conditions; (ii) deeper and more resistive limestone areas, characterised by variable degrees of porosity and clay contents, hence showing more diffuse seasonal variations; (iii) a conductive fractured zone associated with damped seasonal dynamics, while showing a great variability similar to that of the upper layers in response to rainfall events. This study provides detailed images of the sources of drip discharge spots traditionally monitored in caves and aims to support modelling approaches of karst hydrological processes.

## 1 Introduction

Karst regions provide drinking water for a quarter of the world's population (Ford and Williams, 2007; Mangin, 1975). In a changing world, improving the management of vital resources is a key problem, as highlighted in Hartmann et al. (2014). Achieving enhanced management calls for a better understanding of surface and subsurface water movements, known to be strongly heterogeneous in karst areas. The autogenic recharge of the phreatic zone of karst aquifers is driven by water infiltration through the vadose zone (White, 2002). The thickness of this unsaturated zone varies from one karst system to another but is commonly described as two entities: its uppermost layer, the soil joined with to the so-called epikarst which is characterised by high



weathering and porosity of carbonate rocks, overlaying the infiltration zone. The hydrological function of both layers differs from one type of karst to another (e.g. Mediterranean or humid, young or mature karst landscapes; Klimchouk, 2004). While rainfall can directly feed the infiltration zone through sinkholes or open cracks in the epikarst, a part of meteoric water remains delayed in the epikarst (Bakalowicz, 2005). Locally, water can be stored in perched saturated pockets because of strong permeability contrasts with regards to lower layers. Such

epikarst storage was proven to be sustainable enough to host aquatic biota (Sket et al., 2004) or to induce strong dilution of rainwater isotopic signatures (Perrin et al., 2003). In some regions, especially in China, such storage in the subsurface is expected to be great enough to sustainably provide water to populations (Williams, 2008). Nevertheless, these water reservoirs are likely to be seasonally influenced and laterally heterogeneous, interacting with the soil and biosphere through evapotranspiration, while seeping under gravity, or by overflow

after intense rainfall events (Clemens et al., 1999; Goldscheider and Drew, 2007; Sheffer et al., 2011). Such leakage down to the infiltration zone therefore ranges from very slow seepages within the carbonated matrix porosity to quick flows through fractures and cracks in the carbonated rocks (Atkinson, 1977; Smart and Friederich, 1986).

All models describing karst hydrology agree on that dichotomy of matrix and conduits recharge processes

(Hartmann et al., 2014). Karstification is expected to act on the porosity of the bulk rock and therefore on its hydraulic conductivity (Kiraly, 2003). Permanent storage in the vadose zone, responsible for perennial dripping recorded in cave networks, has been confirmed in several case studies (e.g. Arbel et al., 2010). However, compared to the epikarst, the role of the infiltration zone itself in delaying the infiltration and potentially storing groundwater in the matrix porosity remains an open question. In dry periods, dripping with unvarying low

volume discharges are only explained by infiltration via low capacity routes or perched aquifers slowly releasing water into the underlying layers (Smart and Friederich, 1986). Hartmann et al. (2013) modelled the recharge of matrix reservoirs in the vadose zone by lateral exchange with saturated conduits. This confirms the possibility for these processes to occur at several levels within the infiltration zone, making it possible for groundwater to be stored not only in the epikarst, but in several sub-systems of the entire vadose zone.

To support hydrological models, investigation techniques commonly consist of tracer tests or spring flow monitoring, mainly applied to the characterisation of the saturated zone but also tested in the unsaturated zone, to the monitoring of stalactites drip discharge (e.g. Pronk et al., 2009). In particular, such experiments can provide evidence of variable transfer types. Natural caves provide great opportunities to study the vadose zone hydrodynamics from the inside with punctual and direct measurements and/or monitoring. Hydrographs or

hydrochemical monitoring are often a valuable source of information. Although novel promising approaches for building dense cave drip discharge monitoring networks are rising (e.g. Mahmud et al., 2016, 2017), strong heterogeneities of karst areas often make it challenging to build robust networks that adequately capture groundwater storage variations in the vadose zone. Karst subsurface remains poorly known and not often instrumented or monitored. In particular, very little has been achieved to image and monitor perched reservoirs.

Geophysical methods provide non-invasive and integrated tools that can strongly improve karst hydrological knowledge. Hence, numerous studies have been conducted to characterise karst subsurface (see Chalikakis et al.; 2011, for a review). In terms of hydrological monitoring, Valois et al. (2011) and Deville et al. (2013) highlighted the signal of epikarst storage variations in gravity anomalies of repeated gravity measurements. Fores (2016) supported similar measurements with seismic noise monitoring. Recently, Carrière et al. (2016)



successfully used time-lapse Electrical Resistivity Tomography (ERT) and Magnetic Resonance Sounding
(MRS) to identify the role of the porous matrix in regulating water infiltration from epikarst structures,
previously identified by Ground Penetrating Radar (GPR) and ERT surveys in southern France (Carrière et al.,
2013). Meyerhoff et al. (2012) also applied repeated time-lapse ERT measurements to visualize variations in
karst saturated conduits conductivity, assessing the mixing of matrix water and surface water. In parallel,
Kaufmann and Deceuster (2014) have demonstrated the applicability of using ERT to image the porous matrix
associated with karstification processes.

ERT monitoring methods have proved to be highly efficient, especially in hydrogeophysics (e.g. Coscia et al.,
2012; Kuras et al., 2009; Revil et al., 2012) and in engineering and geotechnics for monitoring landslide areas
(e.g. Chambers et al., 2011; Uhlemann et al., 2016), contaminated sites (e.g. Caterina et al., 2017; Kuras et al.,
2016; LaBrecque et al., 1996a) or permafrost regions (e.g. Supper, 2014). The strength of such methods resides
in their effectiveness to track changes in the electrical properties of the subsurface, reflecting variations in
moisture content, groundwater content, temperature or chemical properties. Binley et al. (2015) identify ERT
monitoring as a key technique in the advancing of hydrogeophysical methods applicable for investigating
subsurface processes. A few studies have already used repeated ERT surveys to track hydrological changes in
karst areas, as previously mentioned. They demonstrated the applicability of such techniques with regard to
hydrological purposes in karst, although they spotted real challenges: the heterogeneity of the subsurface making
the interpretation of resistivity models more complex, and the difficulty of practically ensuring proper contacts
for electrodes, especially in presence of outcropping limestone (Chalikakis et al., 2011).

To the best of our knowledge, this paper presents the first attempt of long-term, permanently installed and high
spatial and temporal resolution ERT monitoring of karst subsurface hydrodynamics. Our experiment covers a 3-
year monitoring period of the Rochefort site, a karst area located in South Belgium. The ERT measurements
focus on a 2D profile and comprise two sub-periods: a first 3-month period of daily ERT measurements started
in April 2014 and a second 2-years series of almost uninterrupted measurements from March 2015. Additional
hydrological data such as moisture probes and in-cave percolating water discharge measurements support the
experiment. The monitoring site focuses on a small part of the karst area, at the entrance of Lorette Cave. Such a
local scale approach supports the need to study karst hydrology at all scales (Hartmann, 2016) to build extensive
data sets available for strengthening hydrological models.

## 2    Description of the Rochefort karst system

The study area is located over the central part of Lorette Cave, next to the city of Rochefort in southern Belgium.
Lorette Cave is one of several cavities that belong to the Wamme-Lomme Karst System (WLKS; Marion et al.,
2011), a 10 km long karst area located in the "Calestienne", a band of outcropping Devonian limestone crossing
southern Belgium ENE following the Variscan fold-and-thrust belt (Fig. 1c; Pirson et al., 2008). These units host
the most widespread karsts and caving systems of Belgium (Willems and Ek, 2011). They can be summarised as
two main units: the Charlemont Limestone, that includes four limestone formations, and the Fromelennes
Limestone, at the bottom of which the Flohimont Shales member act as an impervious layer, hydrogeologically
speaking. The Lomme karst system itself results from the crosscutting of the Lomme River, 5 km north-east of
Rochefort, and its main tributary, the Wamme River. The system ends up when the Lomme River meets the
shales and limestones 5 km south-west of Rochefort, at the Eprave resurgence (Fig. 1a).



In Lorette Cave and at a larger scale, in the Rochefort area, limestone layers are part of an overturned syncline
(Fig. 1b) comprising the Charlemont Limestone strata striking N070 with a moderate to high dipping value of
50° to the SSE (Vandycke and Quinif, 2001). All of them are situated within the same lithostratigraphic
formation (the Mont d'Haurs Formation) and form alternating decametric series of well-preserved limestone and
weathered/porous limestone strata with occasional thin clay interbeds.

The study site is part of the Rochefort Cave Laboratory (RCL) (Camelbeeck et al., 2011; Quinif et al., 1997),
located in the central part of Lorette Cave in an underground area that covers about 1 ha at the surface (Fig. 1c).
Most of the area, located at ~225 m above Ordnance Datum (AOD) on a limestone plateau, slopes gently
towards the Lomme Valley, which lies about 165 m AOD. A large sinkhole (typical collapse depression of karst
regions, also called a doline) of ~25 m of diameter and ~20 m deep gives access to Lorette Cave. This cave is
characterised by a well-developed karstic network (Vandycke and Quinif, 2001) comprising large passages with
diameters of several metres that follow the strike direction of the stratigraphic unit (N070), as well as smaller
conduits normal to the main ones (Fig. 1c). The *Val d'Enfer* room, which is in direct connection with the
entrance doline, forms the largest feature of Lorette Cave where several limestone layers outcrop. The
northernmost gallery (the *Fontaine-Bagdad* passage) is another site of interest where structures of the massif are
visible. Most of the galleries of the RCL are located between ~180 and ~190 m AOD, i.e. 40 to 30 m from the
surface.

In terms of hydrogeology, in low water conditions, the water table shows up in Lorette Cave at ~162 m AOD at
the end of a steep small conduit that is about 60 m below the surface of the plateau. A tiny underground river is
also accessible at some points. These accesses to the phreatic zone allow monitoring of the water table level with
CTD (Conductivity Temperature Depth) divers. The Rochefort region experiences mean annual precipitation of
830 mm due to Belgium's temperate maritime climate. Hence, the infiltration reaches its maximum in winter
while evapotranspiration predominates in summer. Heavy rainfall periods, intense storms or snow melting
periods increase the runoff, swelling the rivers, which causes flash floods to occur in the caves of the system.
Flash flood events may temporary rise the saturated zone to a maximum of 174 m AOD (Van Camp et al., 2006;
Watlet et al., 2017) which detrimentally affects the main cavities of the RCL area.

Tectonic activity in Lorette Cave area was described by Vandycke and Quinif (2001) and later monitored by
Camelbeeck et al. (2011), that reported active faults striking N070°E with normal displacement subparallel to the
bedding foliations in Lorette Cave (Fig. 1b). These studies related the strike direction of the faults to the
Hercynian folding and the active movements to the glacial isostatic adjustment.

### 3   Micro-structural observations

#### 3.1   Field, borehole and photoscan surveys

As bedding planes, open joints, fractures and small conduits play an important role in water infiltration in karst
systems, we investigated the local geological structures of the limestone massif. In this area, caves represent
great opportunities to study geological structures from the inside. At the same time, the relatively small scale of
the monitoring area makes it relevant to study in detail the geological structure and lithology of the site.
Observations from multiple sources have been gathered to build a lithological model of the monitoring site,
comprising a field survey and the acquisition of a 3D model of the RCL's main chamber, the *Val d'Enfer* room.



This 3D model results from a drone photoscan of the cavity and allows automatic detection of the orientations of planar structures, and hence a statistical analysis of the main geological structures (i.e. sedimentary layers, fractures/joints and faults) as well as a precise lithostratigraphical log of inaccessible outcrops from the cave's
roof (Triantafyllou et al., 2016). The *Val d'Enfer* room gives a large picture of the limestone layers, on top of which the first electrodes of the ERT profile are attached, in the slope that joins the sinkhole and the cave chamber.

The RCL site was also drilled 2 m away from the centre of the ERT monitoring location (Fig. 1c), just 35 m upstream of the northernmost conduit of the RCL. The 31 m deep borehole was core sampled and surveyed with
downhole well logging. Oriented with respect to the North, it provides useful additional information on the lithostratigraphy and micro-structures of the area monitored with ERT.

Because faults with remarkable rejects are absent in that area, the layers crossed by the borehole, added to those visible in the *Val d'Enfer* room, comprise all the geological strata directly sampled by the ERT measurements. Although the borehole and field surveys in the cave provide direct observations of the geological bodies, they
cannot inform about possible lateral variations in terms of karstification processes and local porosity, likely to vary vertically and laterally in such a karst environment. However, field observations suggest that the lithological nature of each layer observed in the borehole or in the cave is expected to remain constant.

### 3.2   Structural and lithological context of Lorette Cave

Figure 2 summarises the structural observations of the *Val d'Enfer* room (a) and the imaged borehole (c). On the
southern side of the *Val d'Enfer* room, massive limestone layers (50 to 53) correspond to the strata on top of which the ERT monitoring profile is installed in the slope of the sinkhole. A succession of thin clayey limestone (layers 35 to 49) is visible just below. This clay-rich layer is associated with higher percolation discharges than in the rest of the room. This pile of clayey limestone can be simplified as two main clayey layers separated by a more consolidated calcitic limestone, as drawn in Fig. 2c. Underlying layers (34 to 21) show a remarkable
homogeneity in terms of lithology or weathering rate. They correspond to the first layers crosscut by the borehole, where the first 3.4m core samples, compact in the cave outcrop, exhibit higher weathering rates, porosity and fracture intensity, typical of an epikarst layer. Deeper in the borehole, the underlying 7.5m (layers 20 to 16) are mostly massive poorly weathered limestone crosscut by only few joints. The area between 11 and 16m deep (layers 15 to 13) is however characterised by a high number of joints and sedimentary
discontinuities/beddings partially to totally karstified; also evidenced by an increased porosity.

Constraining the geometry of discontinuities (i.e. joints and sedimentary beddings) is crucial to understand the dynamic of the local water infiltration. Some of the encountered open joints are 2 to 3 cm wide. The openings with dip direction N160 and dip values around 50° mainly represent bedding planes (S0). Based on their geometry, three groups of joints orientations have been identified on the borehole and in the cave (see
stereograms and rose diagrams in Fig. 2b-d). The first one comprises planes striking N070 strongly dipping (~50°) to the south. They consist of joints and few faulted structures subparallel to the S0, similar to the active fault network evidenced by Vandycke and Quinif (2001). As shown by this study, these faults are marked by downdip slickenlines (Fig. 2a) that mark a normal displacement. A second major joint orientation is marked by comparable striking values (N070) but with high dips (~60°) to the north. In the *Val d'Enfer*, downdip
slickenlines have been recognised on these faults showing normal displacement. The normal kinematics are also





attested on the borehole log showing displaced S0 layers on both parts of few N070-N60 faults. The latter could represent conjugated faults to the direction of the first fault reported above but need additional microstructural surveys. One particular fracture with that orientation is also visible in the northernmost gallery of Lorette Cave, and can be followed along 25 m parallel to the gallery, with an opening of 4-5 cm. A third joints subset shows

subvertical joints mainly represented by a mean strike around N150 and few conjugated joints striking N330 (Fig. 2).

In the 2D geological model of Fig. 2c, two main open fractures with respective orientation N070-N60 and N300-NE80 identified in the porous layers (15 to 13) are extrapolated to the surface, as they may play a major role in the water percolation.

**4   Environmental monitoring**

At the surface of the RCL site, a small building, located at the border of the large sinkhole, hosts the instruments data loggers and the resistivity meter. The Eastern part of the site is mostly asphalted, with a parking area and two minor roads, while the rest of the area is wooded, including the sinkhole where the ERT profile is installed (Fig. 1c).

**4.1 Sensor network installation**

Several environmental sensors have been installed at the RCL site. First, a vertical profile of five water content reflectometers (WCR) from Campbell Instruments (CS616) are installed 2 m away from the ERT monitoring profile. They provide data with a resolution of 0.1 % in volumetric water content (VWC) and a sensor variability of 0.5 % and 1.5 % VWC in dry and humid conditions respectively. The probes are inserted at depths of 10, 30,

50, 75 and 105 cm, and have been operational since May 2015. Their sampling rate was 1 hour for the first months of measurement and was changed to 1 minute afterwards. As the average soil thickness is only 40 cm at the RCL site, a portion of fractured and weathered limestone mixed with clays and roots needed to be excavated to 105 cm. This material was replaced after each WCR was installed. They are therefore surrounded by a mixture of limestone blocks clays and soil materials. Such heterogeneous materials make the calibration of the WCRs

rather challenging; the porosity ($\varphi$) of the soil and rocks surrounding each probe being hard to assess. Humidity data will therefore mostly inform about the dynamics of the infiltration at the location of the vertical profile. Some assumptions on the porosity around the probes based on maximum thresholds reached during the monitoring period could also be proposed to estimate the saturation ($S = \text{VWC}/\varphi$).

Additionally, Lorette Cave is equipped with percolation discharge monitoring concentrated in three specific

locations. Two drip discharge gauges are installed in the *Val d'Enfer* room, one of which (PWD1) monitors flows dripping through a subvertical open fracture oriented N160 in an clayey limestone layer, the other one (PWD2) being installed under a karstified area where drips come out of a particularly porous limestone layer. The third station (PWD3) monitors one stalactite built on a massive limestone layer associated with very slow discharge in the northernmost passage at the vertical of the ERT profile. This area is generally much drier than

the *Val d'Enfer* room. The thickness between the surface and the monitored inlet flows is ~25 m for PWD1 and PWD2 and ~33 m for PWD3. PWD1 and PWD3 have been monitored since 2001, but PWD1 suffered from instrumental problems from 2013 to 2015, when the instrument was replaced. The complete network was





finalised in March 2016 with the additional PWD3 installed in the framework of this study. Measuring drip discharge is usually complex as calcite deposits can perturb the instruments, while the great variability of flow

regimes is particularly challenging for the instrumental design. PWD1 and PWD2 are made of an auto-siphoning gauge with capacitive sensors designed by University of Mons, based on an original prototype from the Royal Observatory of Belgium (Kaufmann et al., 2016). The dripping water is collected in an inverted cone feeding a small upper tank which in turn feeds a larger lower tank. Capacitive sensors plunge in each tank and return high frequency FM signals. This sampling allows the emptyings of the tanks to be counted to estimate the flow rates.

Using a small and a large tank increases the range of flows supported by the system (0.5 to 100 liters/hour). Since time resolution depends on the discharge, an interpolation is required to get a constant time-step of 10 minutes. The PWD3 instrument only comprises one capacitive sensor surrounding the tip of the monitored stalactite. The growing water drop creates a decrease of the FM signal, followed by a sharp increase triggered by the drop's fall.

Specific electrical conductivity (SpC) measurements are also performed in-cave at the PWD1 monitoring station, as well as at surface for monitoring rainwater conductivity, next to the ERT profile. Both measurements are performed using a Campbell CS547A probe (accuracy of ±5%).

Rainfall was monitored for the whole period of ERT monitoring using a Luft tipping bucket type rain gauge with a 1 minute sample rate, located on the RCL site itself. The locations of the WCR profile and rain gauge are

shown in Fig. 1c. Additional Potential Evapotranspiration (PET) data are also available. PET is derived from the Penman-Monthei relationship (Allen et al., 1998) based on data from a meteorological station (Pameseb) located 5 km from the Rochefort monitoring site.

### 4.2    Environmental data results

Figure 3 shows the rainfall and PET data (a) in comparison with the soil moisture recorded by the WCR (b). This

gives an overview of the climatic conditions experienced during the ERT monitoring experiment. The year 2015 can be considered as normal in terms of weather conditions, with rainfall homogenously distributed, except for a short dry period in September 2015. PET also follows expected trends, resulting in negative values of effective rainfalls during summer (given by rainfall – PET). In comparison, 2014 and 2016 were more unusual, experiencing particularly wet summers, leading to very few periods of negative effective rainfall. This means

that at least the uppermost layer was continuously fed by rainwater, which should result in high average moisture contents, as was monitored by the WCR for the 2016 period. However, a remarkable dry period affected Southern Belgium at the end of summer 2016 (from August 6th to October 15th), with only 53 mm of precipitation. This is extremely low compared to the seasonal average for the area which normally equals 172 mm for the same period of time, based on the seasonal averages provided by the Belgian Royal Meteorological

Institute. This period will be particularly interesting to look at with the ERT monitoring, as the lowest VWC measured at site was reached in the top layers. Particularly high resistivity values are to be expected in the surface layer during that period. Especially, comparing ERT data from summer 2015 and end of summer 2016 will be useful in identifying the role of PET and rainfall in the moisture contents of deeper layers.

Overall, soil moisture data show repeated rainfall infiltration processes. Every significant precipitation event

progressively infiltrates the soil layer producing a sharp increase of VWC followed by an exponential recession curve. The delay between the beginning of the rainfall event and the first arrival of infiltrating water depends on



the intensity of the rainfall event, evapotranspiration conditions, the depth of the moisture probe and hydraulic conductivity parameters defining the soil retention curve. In winter, a delay of ~14 hr is observed between the 10 and 105 cm deep probe. In summer, this delay can be significantly longer, up to several days. The influence of

evapotranspiration is clearly noticeable as fewer peaks are present in the VWC data set. After long droughts, such as that of August and September 2016, a delay of 85 days is noticed between the first moisture content peak observed at 10 cm depth and that observed at 105cm.

In parallel, in-cave percolating water discharge data bring crucial information on the infiltration processes occurring in the vadose zone at the RCL site. The three stations show different discharge dynamics given their

location and the type of inlet flow that they sample. Smart and Friederich (1986) developed a drip discharge classification based on the relationships between maximum discharges and coefficients of variation of the discharge, they can be described as vadose flows for PWD1, PWD2 and seepage flow for PWD3 respectively. Vadose flows refer to high discharges, albeit lower than for shaft flows, with a high variability, especially regarding their rainfall events responses. Seepage flows exhibit significantly lower discharges with low

coefficients of variation but noticeable seasonal changes. Despite being classified as vadose flow, the PWD2 data set exhibits a strong seasonal pattern. It actually samples more of a dripping zone rather than one single inlet flow associated with one stalagmite or fracture. This could lead to overestimating the maximum discharge regarding the approach of Smart and Friederich (1986). PWD2 could therefore be described as seasonal drip, which differ from seepage flows by their higher coefficient of variation, following the modification of the

classification after Baker (1997). The closeness of PWD1 and PWD2 exhibiting different discharge regimes testifies to the high heterogeneity of the Rochefort karst.

Seasonal cycles affecting percolating water are strongly related to effective rainfall and soil moisture data, as shown in Fig. 3. Similar observations have been described and analysed in multiple studies, highlighting the buffering role of the epikarst in water infiltration (e.g. Genty and Deflandre, 1998; Poulain et al., 2015; Sheffer

et al., 2011; Aquilina et al., 2003). Arbel et al. (2010) distinguish perennial drip discharge, which explains the bottom threshold visible in PWD2 and PWD3 in summer (Fig. 3c), and seasonal drips that stop during summer and are characterised by longer recession times. Additionally, post-storm drips directly follow rainfall events and decay after a few weeks, exhibiting a high discharge variability. The first two types can be related to diffuse flow that propagates through the matrix, while the last type refers to quickflows and conduit infiltrations (Hartmann et

al., 2014; Lange et al., 2010; Perrin et al., 2003). Overall, these classifications highlight the duality of water infiltration and recharge in karst systems.

Unlike PWD2 and PWD3, PWD1 does not exhibit a clear seasonal trend, even though the baseflow threshold and post-storm drip decrease during driest periods, while longer recession curves are observed. This is especially the case for the August and September 2016 drought. Poulain et al. (2017) provides a specific analysis of the

diffuse flow and quickflow components of PWD1, supported by a vadose dye tracing test. It confirms the two-flow regime as a mixing of matrix and conduit infiltration. PWD2 and PWD3 seem to depend more on diffuse flow through the matrix but a part of quickflow is still present in the signal. In conclusion, drip discharge data reflect well their station's location: PWD1 samples inlet flows from an open fracture crosscutting a clayey limestone layer, which explains the great quickflow component from post-storm drip type percolation through

the fracture. PWD2 monitors drip discharge from a porous limestone layer; water coming out directly from the rock matrix, without the presence of stalactites. Finally, PWD3 is installed on a dry location and samples inlet



flows from a stalactite built on a massive limestone layer. This explains its very low observed perennial drip discharge described as seepage flow.

The electrical conductivity of the percolating water (Fig. 3d) displays some variations following rainfall events
and related recharge processes, but no seasonal trends are evidenced. The observed values average 0.25 mS/cm (40 Ωm), while maximum values of 0.33 mS/cm (30 Ωm) are recorded after long droughts. Rainfall events result in rapid decreases of the electrical conductivity which sometimes dip to 0.15 mS/cm (65 Ωm). These values are attributed to rain water rapidly infiltrating conduits, mixed with more conductive groundwater. The 0.44 mS/cm threshold is believed to account for the pore-water maximum electrical conductivity in the subsurface of the
RCL site, whereas recharge processes due to rainfall tend to result in decreased electrical conductivity. Electrical conductivity of the rain water is also monitored at the surface ranging from 0.20 to 0.04 mS/cm (50 to 250 Ωm). Given the very small delay between rainfall events and in-cave discharge increases, the significant difference between the minimum electrical conductivity measured for rain water (0.04 mS/cm) and dripping water (0.15 mS/cm) testifies to rapid mixing processes with groundwater and/or efficient ionic leaching by the percolation
water. Such rapid changes are in accordance with findings of Hunkeler and Mudry (2007). Poulain et al. (2017) also provide a study of the relationship between the discharge flows and the electrical conductivity at the RCL site.

## 5   ERT Monitoring

### 5.1   ERT monitoring installation

Seven preliminary ERT surveys around the RCL site were conducted in 2013. They resulted in identifying the sinkhole giving access to RCL as an area with heterogeneous electrical resistivity features likely to be of interest for monitoring complex hydrological processes. A profile of 48 electrodes, with a pronounced topography, was therefore installed permanently. Twenty-eight electrodes from this profile are set at the top of the limestone massif and 20 others along the slope of the sinkhole (Fig. 1c). Most of the electrodes are buried 20 to 30cm
below the surface and made of stainless steel hollow tubes with diameters of 2cm and lengths of 12cm (for total surface of contact of ~150cm²) (see photos in Fig. A1). Good electrical contact of each tube with the soil was ensured with bentonite. Because a limestone stratum is outcropping at the bottom of the sinkhole, the 5 southernmost electrodes of the profile are directly bolted into the rock. A stainless steel wedge anchor was used to fix a 100cm² stainless steel plate to the rock (total surface of ~110cm²). A protective cap made of polyurethane
foam covers each electrode in order to reduce corrosion processes and for safety reasons. The electrode spacing was chosen to be 1m, as recommended amongst others by Clément et al. (2009) when monitoring shallow recharge processes.

Two acquisition systems were installed at the RCL site. A first testing period lasted from March 2014 until June 2014. An Automated Time-Lapse Electrical Resistivity Tomography (ALERT) acquisition system developed by
the British Geological Survey (Kuras et al., 2009) collected daily dipole-dipole (DD) measurements. After this testing period, we installed a 4-channel Iris Syscal Pro resistivity meter in March 2015, which is still presently measuring. Daily multiple gradient (GD) and DD data were collected, except for summer 2015 and winter 2016 where DD arrays, which require higher injection power, suffered from battery malfunction issues. Both acquisition systems are remotely controlled from the office. Data are automatically sent to a server and checked





for measurement errors. The acquisition system is installed in a brick shelter, furnished with a wired internet
connexion and a 230 VAC power access, providing ideal infrastructure for an ERT monitoring site.

The measurements protocols involve DD and GD types. They were chosen because of their effectiveness for
multichannel data acquisition purposes as well as their good image resolution capabilities (Dahlin and Zhou,
2004). DD type surveys use dipole lengths (*a*-factor) of 1 to 3m and dipole separation (*n*-factor) of 1*a* to 10*a*,

and involve reciprocal measurements. Exchanging current and potential electrodes should ideally deliver the
same results, as stated by the reciprocity theorem (Parasnis, 1988). Comparing forward and reciprocal
measurements provides a robust method for estimating the data error and quality (LaBrecque et al., 1996b;
Wilkinson et al., 2012). GD type surveys use a combination of dipole separation (*a*-factor) of 1 to 4m and a
current-electrode separation (*s*-factor) of 1a to 4a (for further information on multiple-gradient arrays, see Dahlin

and Zhou, 2006). Although this configuration provides a lower depth of investigation than that of the chosen DD
array, it has an improved resolution for shallow depth. Because of the non-symmetric electrode configuration for
GD arrays, reciprocal measurements are practically unsuitable for multichannel data acquisition. This can lead to
particularly long acquisition duration time, which increases the risk of real changes occurring during a
measurement sequence. Hence, acquiring daily reciprocal GD measurements would be too time consuming; only

normal GD measurements were performed for time-lapse monitoring.

### 5.2    ERT data processing

#### 5.2.1    Data processing, quality control and error estimation

We developed a semi-automated workflow involving routines for data acquisition, storage, filtering, inversion,
and visualization. A first data filtering is applied on the repeatability error of each measurement. During the

acquisition, the potential difference on the measurement dipole of each quadrupole is measured two to four times
by the resistivity meter. Distributions of the repeatability error are shown in Fig. 4a. For DD arrays, data having
repeatability with a standard deviation (repeatability or stacking error) over 5%, as well as measured potentials
lower than 1 mV, are automatically filtered. Following this step, reciprocal errors are computed for the DD type
dataset.

Reciprocal error is the resistance difference between normal and reciprocal measurement, i.e. when current
injection and potential dipoles are swapped. Figure 4b shows the distribution of the reciprocal errors for the
whole DD type dataset, after filtering for repeatability errors and low potentials. Data with relative reciprocal
errors over 20% were also removed. Reciprocal errors are used as a noise estimate for the inversion procedure,
where the resistance of each measurement needs to be weighted. Overall, filtering on repeatability error and

reciprocal error leads to 15% of all the DD measurements being rejected, mainly due to too low measured
potentials. GD type surveys have no reciprocal measurement available for each daily dataset. A punctual
reciprocity test was performed on GD arrays and showed relative reciprocal errors slightly higher than those of
DD arrays. This is attributed to the fact that GD surveys have a measured resistances range significantly broader
than that of the DD surveys. Furthermore, the signal to noise ratio shows a slightly different order of magnitude

between normal and reciprocal measurements in case of GD surveys, which is expected to increase the reciprocal
error. Another possible explanation for this is that real changes may occur during the GD surveys. Since the GD
reciprocals take longer to measure (single channel) than the DD reciprocals, there is more time for greater
changes to occur, leading to greater differences between forward and reciprocal measurements. This also



explains why the DD reciprocal errors are greater than the DD repeatability error: stacking measurements are
measured close together in time, but forward and reciprocal pairs are separated by larger times. Given that GD
arrays have no reciprocal measurements available for the entire monitoring period, the repeatability error
filtering threshold was set down to 0.5%, which also takes into account the lower mean of the repeatability error
distribution compared to that of the DD arrays, as visible in Fig. 4a. Following this filtering, only 1.5% of GD
measurements were rejected.

Contact resistances along the ERT profile also showed high temporal variations, following the moisture
conditions at site (Fig. 5a). The high clay content of the soil at RCL ensures a very good electrode/ground
contact in humid conditions. It however favours shrinking during dry periods that can reduce the surface contact
of electrodes with surrounding soil materials. Such processes therefore increase the contact resistances that, in
turn, are a source of increased measurement errors. Higher contact resistances are usually noticed as they
produce greater repeatability errors and reciprocal errors. In dry periods, this leads to a higher number of rejected
data after filtering. In August 2016, electrode #12, placed in the middle of the slope of the sinkhole started to
show significantly bad contact resistances (> 50 kΩ) which induced poor measurements quality. For time-lapse
processing, all the quadrupoles comprising electrode #12 were therefore continuously rejected, which reduced
the maximum number of measurements per survey and the resolution in that part of the ERT profile. Full surveys
are composed of 990 DD reciprocal measurements (901 without electrode #12) and 1420 GD measurements
(1296 without electrode #12). Figure 5b and c summarise the percentage of rejected measurements per survey for
the monitoring period. Days with more than 10% of rejected data were removed from the time-lapse dataset, and
are therefore not shown in Fig. 5b-c. Given the greater amount of rejected DD measurements, this results in 467
DD and 588 GD data sets.

In December 2015, despite the fact that the monitoring site was equipped with AC power supply, the injection
batteries started to fail because of the increased power demanded by the resistivity meter to deliver higher
voltages, especially for DD surveys. The battery malfunction led to the rejection of almost all the DD surveys
acquired during that period. GD surveys were less sensitive to the problem because the power and voltage
requirements were lower on average. The greater percentage of rejected data in 2015 is also attributed to the
batteries being slightly less efficient for repeated high voltage injections. The original starting type batteries were
replaced by deep-cycle gel batteries in July 2016, especially designed for deep discharge, similar to those
frequently used with solar panels. The long drying period from August to October 2016 is however responsible
for a progressive increase in the amount of rejected data for both DD and GD surveys.

Temperature variations of the subsurface can have significant impacts on resistivity data (Brunet et al., 2010).
Because a marked temperature gradient is expected in the sinkhole, we modelled the 2D temperature field using
the framework of pyGIMLI (Rücker et al., 2016; www.pygimli.org) using data from a network of 8 temperature
probes installed along the ERT profile and a 105 cm vertical profile of 5 temperature probes. Correcting for the
effect of temperature in time-lapse ERT can be implemented in several ways. We chose to apply the method
described in Hayley (2007) in which both the data and the modelled resistivites are corrected for temperature
effects.



### 5.2.2    Time-lapse inversion

Data are inverted with Boundless Electrical Tomography software (BERT), which is based on a finite element modelling (Günther et al., 2006; Rücker et al., 2006). Each of the DD and GD data sets after filtering and correction for temperature effects constitutes one data set for the inversion, whereas the inverted results of the

first data set of the DD and GD series constitute the reference model for the whole time series. The inversion is carried out using a $L_1$ (robust) data constraint to mitigate bias produced by outlying data on the convergence of the inversion and a $L_2$ (smoothness) model constraint. The time-lapse procedure comprises an additional time regularisation constraint ($\lambda_t$=50) for the inversion of each subsequent data set, linking each of the inverted models to the reference model. This is used to smooth over time the inverted results between each other, while

facilitating the convergence of each inversion. Note that the structural assumptions presented in Sect. 3 are not used as model constraints in the inversion. This choice was made because of the uncertainty concerning the spatial continuity of structural observations in such a heterogeneous karst context. Time-lapse inversion achieved acceptable convergence between the observed data and the reconstructed model data as average RMS misfits of 11.7% (standard deviation 1.2%) and 2.4% (standard deviation 0.2%) for DD and GD arrays were respectively

retrieved. Higher RMS errors of DD arrays are not really surprising. They are attributed to the lower signal to noise ratios of dipole-dipole arrays and the greater number of large dipoles configurations in the chosen DD protocol. Chi-squared is another way for evaluating the convergence of the inversion. A chi-squared value about 1 generally tells that the error model is appropriately calibrated and does not lead to data underfitting or overfitting. Time-lapse inversion resulted in chi-squared values of 0.87 (standard deviation 0.1) and 0.84

(standard deviation 0.1) for DD and GD time series respectively.
BERT provides a coverage parameter calculated from the sum of the absolute sensitivities of the measurements. Areas with high coverage are better constrained by the measurements and the modelling choices than low coverage areas. The coverage is usually incorporated in the data visualisation as a transparency mask that is used to weight the data depending on their associated coverage values.

## 6    ERT imaging results

### 6.1    Resistivity distribution

The inverted resistivity images of DD and GD arrays show a great dispersion. Images from DD arrays have a better coverage at depth than those of GD arrays. Hence, they are able to identify deeper structures, while GD arrays have a better resolution at shallow depth. Figure 6a presents an image after inversion of the DD array of

May 2015, which corresponds to a period of average weather conditions. A 1 to 3m thick, highly conductive layer (<200 Ωm) is present at the surface of the limestone plateau (A in Fig. 6a). This layer progressively disappears along the slope of the sinkhole. The northern half of the ERT image also shows a highly resistive (>3000 Ωm) area below the surface layer (B). A second conductive zone (C), dipping ~70° to the North, is revealed in the central part of the image, and corresponds with the topography inflexion. Resistivity values of

that area are comparable with those of the surface layer. To the South, additional thin conductive layers appear at the surface (D) and 3 to 5 m below the surface of the slope (E), under a more resistive thin layer. The rest of the image shows high resistivities up to 5000 Ωm. Results of the GD array (Fig. 6b) for the same period also image the conductive layer at the surface, the high resistivity area in the northern part of the survey, and the thin



conductive layer below a thin more resistive layer in the slope. The highly conductive zone dipping steeply to the north in the middle of the DD image result is however not reconstructed at depth on this section. This is not surprising as configurations of the GD protocol leads to lower sensitivities at depth.

Field observations corroborate the presence of the conductive layer on the plateau, as this area is characterised by a ~40cm clay-rich soil layer, as previously mentioned. This layer becomes progressively thinner in the slope of the doline, and disappears at the sixth electrode from the bottom, which required the first six electrodes to be attached to the outcropping rock, as described in Sect. 5. The conductive layer at the top of the plateau is however thicker than 30cm. Even if a blurring effect is expected from the smoothness constraints, this layer seems to be about 1 to 3m thick. It is interpreted as the soil and epikarst, where the porosity is higher than in deeper limestone layers.

The high resistivity values correspond to those of reasonably non-karstified limestone. Intermediate values may indicate to karstified limestone with a low clay or moisture content. Such values could also result from clay rich interbeds between limestone strata. Given the resolution of the ERT image, such thin features cannot be precisely delineated. In the case of a highly conductive thin shape surrounded by highly resistive materials, the inversion process could only build a larger conductive area with an intermediate resistivity value. The strongly dipping conductive feature that crosses the resistivity image in the middle also deserves discussion. It could represent a fractured zone either filled with clays or characterised by a higher moisture content than the surrounding materials. The time-lapse imaging will give some insights on the role of this part of the section in hydrogeological processes taking place in the near surface by highlighting changes in resistivity through the year.

### 6.2    Time-lapse imaging

There are several ways to visualize spatio-temporal changes in resistivity resulting from a time-lapse ERT inversion, such as computing resistivity ratios, log of resistivity ratios or percentage of change in resistivity or log of resistivity. The basis on which the time-lapse comparison is made must also be appropriately assessed. It may involve visualisation of resistivity variations that occur between each data set, highlighting sharp resistivity changes. Lower frequency resistivity variations usually need to be visualized as changes with respect to a baseline. Such a baseline does not necessarily have to be the same as the reference model used in the time-lapse inversion process. In many studies, the baseline used to track resistivity changes consists of resistivity measurements performed before the process to be monitored begins. In some cases, this stage is easy to assess. For instance, when monitoring artificial water injection, the baseline commonly consists of an ERT survey made prior to the start of the injection (e.g. Robert et al., 2012). In other cases, including this one, a clear baseline is more difficult to assess because the beginning of the process of interest, i.e. recharge processes, is not clearly identifiable. Here, the seasonality of the vadose zone moisture conditions (Fig. 3) complicates the choice of a clear baseline to visualise the time-lapse resistivity images.

As this study focuses on groundwater recharge, a baseline in dry conditions was chosen. The long drought of August to mid-October 2016 during which WCR moisture measurements dropped dramatically (Fig. 3) corresponds to the driest conditions encountered during the monitoring period. ERT surveys of October 15[th] 2016 corresponding to the last day of this drought were therefore chosen as baselines ($\rho_0$). Consequently, change in log of resistivity with respect to that baseline is computed (Fig. 7b) for each data set ($\rho_t$) as:





$$\Delta\rho = \left(\frac{log\rho_0}{log\rho_i} - 1\right) * 100 \qquad (1)$$

Our preference was for change in log of resistivity because it gives a better overview of gradual spatial variation than changes in absolute resistivity.

Figure 7b shows changes in log resistivity for 10 of the 467 DD data sets. Although this kind of display highlights broad variations in resistivity, it assumes that the nature of the underground materials would give comparable resistivity changes if exposed to similar moisture conditions. This is usually the case in fairly homogeneous environments, whereas it is not necessarily valid in highly heterogeneous contexts. Such visualisation could therefore overlook subtle changes in regions associated with resistivity variations of smaller

amplitudes. To address that problem, our approach consists of displaying images of normalised resistivity ($\rho_{norm}$) rescaled between 0 and 1 (Fig. 7c), such as for each cell (*c*) of the mesh:

$$\rho_{norm}^c = \frac{\rho_i^c - \rho_{min}^c}{\rho_{max}^c - \rho_{min}^c} \qquad (2)$$

Where $\rho_i^c$ corresponds to the resistivity of a cell at a time step *i*, while $\rho_{max}^c$ and $\rho_{min}^c$ are respectively the minimum and maximum resistivity value of each cell over the whole time series. In this way it is possible to interpret the resistivity of each cell with reference to its variation in the considered timespan.

Overall, changes in log resistivity compared to the driest conditions encountered at the surface during the whole monitoring period show that the majority of the variation in resistivity is negative. This means that most of the subsurface experienced a decrease in resistivity, consistent with an increase in the moisture. The area most affected by intense (>10%) decreases in resistivity is the surface of the plateau. A surface layered structure (~2.5m deep) can be seen to vary temporally with higher amplitudes than the rest of the model. It corresponds to

the conductive layer displayed in Fig. 6. Highest negative changes in resistivity are reached during winter (December to March). The basis of this layer does not follow perfectly the surface topography, going a little bit deeper at the North and South of the plateau. Especially, in the middle of the profile, in the southern side of the plateau, a small area (~4m in extent) remains at very negative values for a longer time. This area seems therefore to act as a small basin in terms of ground moisture, highlighting the very high spatial heterogeneity expected in a

karstic medium.

Below this surface layer, changes in log resistivity progressively attenuate, in the area of the higher resistivities imaged in Fig. 6, except for a circular shape discernible in the middle part of the plateau (centred around 5m deep). This area that corresponds with the lower resistivity anomaly in Fig. 6 also experiences high negative resistivity variations in winter while showing slight increases in resistivity in summer compared to the driest

condition at the surface. Such a behaviour in summer is also noticeable in deeper parts of the sinkhole area that are associated with low resistivity values. However, in winter resistivity changes remain less negative in that area. The changes in the surface layer of the doline follow the same dynamic as that of the surface of the plateau, except that it shows less intense log resistivity decrease (<10%). The feature strongly dipping to the south, starting at the inflexion of the slope of the sinkhole (associated with high conductivity in Fig. 6), also has a

dynamic in terms of resistivity changes discernible from its surrounding areas. It shows mostly negative variations, smaller than those of the surface layer.

Figure 7c comprises images of the normalised resistivities for the same time steps as those of Fig. 7b. Besides providing images with an enhanced contrast, it allows clarification of the temporal signature of some regions of the image, especially the deeper ones. For example, the strongly dipping feature has a relatively low resistivity in



the image of 24$^{th}$ July 2015 (Fig. 7b-3) that equals negative changes in surface layers of both the slope of the doline and the plateau. When comparing that information with the normalised resistivity image (Fig. 7c-3), it appears that this strongly dipping feature displays normalised resistivity values (~0.2) closer to their measured minimum than those of the surface layers (~0.6). This means that for this time step, the dipping feature is very close to the maximum measured negative resistivity change while surface layers are in average conditions with respect to the resistivity distribution observed during the monitoring period. Similar observations can be made for the deep circular feature in the middle of the plateau for time steps of 18$^{th}$ August 2016 and 1$^{st}$ October 2016 (Fig. 7b-8 and b-9) where negative variations of resistivity similar to some of the surface layers and surrounding areas are not equally represented by the normalised resistivity images (Fig. 7c-8 and c-9). The deep area of the slope of the doline is also interesting to look at in time steps of 1$^{st}$ June 2015, 8$^{th}$ August 2016 and 18$^{th}$ August 2016 (Fig. 7b-2, b-7 and b-8), as little change in resistivity relative to the minimum status can be seen in terms of normalised resistivity (Fig. 7c-2, c-7 and c-8).

In conclusion, this highlights the fact that several subsurface regions of the ERT profile experience pronounced changes in resistivity through the monitoring experiment period. These regions have their own resistivity signatures, as illustrated in Fig. 6, which testifies to the substantial heterogeneity inherent to karst systems. To simplify the observations, Fig. 8 clusters the recovered resistivities of the subsurface of the ERT profile in eight distinct regions that display different spatial and temporal resistivity signatures, based on their average resistivity values and arbitrary thresholds. It differentiates surface and subsurface layers (1, 2, 5 in Fig. 8) and deep regions (3, 4, 6, 7, 8 in Fig. 8). Interestingly, these regions display different dynamics in terms of the temporal resistivity evolution in Fig. 7. Especially, dynamics of the surface layers look dissociated from those of three deeper regions: the moderately resistive circular feature in the middle of the plateau (3), the deep layer in the slope of the doline area (7) and the high dipping feature (8). The rest of the image does not show much noticeable change (6) except low variations in the rest of the deep parts of the plateau in the right side of the image (4).

## 7    Discussion

### 7.1    Linking resistivity distribution to lithology and microstructures

The highest recovered resistivities are around 6000 Ωm, which is a typical signature of low-porosity limestone. Other regions show lower resistivity values that range from 2000-3000 Ωm for anomalies 3 and 7 of Fig. 8 to ~500 Ωm for anomaly 8. The surface layers (1 and 5) also display very low resistivities close to 500 Ωm on average, which gradually increase with depth up to 1000-2000 Ωm for the subsurface layer (2). With the karst environment in mind, there are two main explanations for such conductive anomalies: a greater clay versus limestone ratio, or a higher moisture content. On the first hand, a clay-rich composition is already observed in the very thin soil layer. Bedding planes and large open fractures could also be filled with clays or a mixture of clays and percolating water. On the other hand, if conductive anomalies are to be explained with higher moisture contents, they reveal at the same time areas with higher porosity. It is therefore very likely, and hence not surprising, that the ERT profile samples subsurface regions with different degrees of porosity. It remains to be determined whether such increased porosities would account for matrix or conduit porosity types, both frequent in karst subsurface.





The structural model presented in Fig. 2 is an important source of information in order to further investigate the spatial resistivity distribution. It can be summarised as a series of massive limestone strata that includes a pile of clayey limestone layers next to the slope of the doline, porous limestone strata with a greater fracture intensity in the middle, and massive limestone layers interbedded with three thin porous limestone strata in the northern part of the ERT profile (Fig. 2). All these strata are dipping ~55° to the South. Several largely opened fractures are also visible in the borehole image, two of the most important ones cutting across the inflection point of the profile's topography when extrapolating their ~65°N dip. They are representative of one of the preferential fracture directions evidenced in the cave. Finally, the top layer hosts the thin soil layer and a ~3m porous, fractured and weathered layer that refers to epikarst features, as highlighted by the core samples.

ERT forward modelling provides a useful tool to verify structural hypotheses, to test their electrical resistivity response and hence to guide the interpretation. In an attempt to account for both the structural information (Fig. 2c) and the ERT results (Fig. 6), the karst underground has been segmented in six resistivity regions (Fig. 9a). The lithological pieces of information are converted into resistivity values using assumptions chosen to best fit the observed data. The fractured area (< 1m wide) is given a very conductive value of 200 Ωm, to simulate high clay and moisture contents. A value of 600 Ωm is given to the clayey limestone layers, while an average value of 3000 Ωm represents a porous limestone layer. A lower resistivity value is also given to the first limestone layer in the flank of the doline, as a high degree of weathering is observed in that area. The thin soil layer is also represented as a 800 Ωm layer. These values are chosen to compare with field data acquired in average climate conditions, e.g. not too humid or too dry periods. The epikarst layer is deliberately omitted on the plateau's side because it is likely to be spatially heterogeneous.

Based on the resistivity model, the potential of each quadrupole of a given protocol is computed by forward modelling, resulting in a synthetic apparent resistivity data set. This synthetic data set can therefore be inverted and the resulting ERT model can be compared to that produced with observed data. Synthetic apparent resistivities are computed using BERT forward model for full DD and GD protocols (i.e. without removing electrode #12) together with a randomly distributed noise level of 5%. The synthetic data sets are inverted afterwards with an error model defined as the mean error distribution used for the inversion of the whole DD and GD field data time series.

Figures 9b and c present the results of the DD and GD protocols respectively. The first remark is that these reconstructed images look very similar to the inverted field results presented in Fig. 6. The sensitivity of the GD protocol does not allow proper recovery of the fracture anomaly while the DD survey successfully images it. Nonetheless, the thickness and the original resistivity value (200 Ωm) of this fracture area is not properly recovered, which is most likely due to the fact that the thickness of the anomaly is smaller than the electrode inter-distance and model cells sizes. Moreover, the inversion uses a smoothness constraint in the regularisation (Constable et al., 1987) which tends to blur the boundaries between areas with different resistivity values. Using structural constraints on the ERT inversion could have mitigated such effects. However, as stated in sect. 5.2.3, we chose not to include such constraints to avoid introducing biases.

More interestingly, with a model value of 600 Ωm, the clayey limestone layer is also clearly imaged after inversion of the synthetic data set. In this case, both the thickness and the resistivity value set in the theoretical model are successfully retrieved, albeit with a progressive resistivity gradient at the boundary with massive limestone layers. To explain the moderate resistive circular anomaly in the middle of the plateau, a value of 3000



Ωm was chosen for the porous limestone. This leads to a poor reconstruction of this layer, resulting in a ~8m deep circular anomaly surrounded by high resistivities accounting for the massive limestone regions. This is both attributed to the lack of resolution at greater depth and the lower contrast in resistivities between the altered and

the massive limestones. In any case, such a circular anomaly compares well with that identified in the inversion of field data of the DD surveys and to a lesser extent of the GD surveys. This means that this circular anomaly can successfully be explained as a limestone layer with lower resistivity than surrounding massive limestone strata, for which the low imaging resolution at depth results in averaging the resistivity with that of its surroundings, i.e. the massive limestone.

Overall, without taking into account an epikarst layer that is thought to explain the small irregular conductive anomalies in the northern side of the inverted field data results, the inversion of these synthetic data sets explains the resistivity distribution substantially. Pearson's correlation coefficients of 0.74 and 0.79, for the DD and GD surveys respectively, were also computed to compare these synthetic models with inverted models of March 2017 (Fig. 6), representative of average moisture conditions at the RCL site. Such correlation coefficients

support the visual similarities between the synthetic and observed images. In the upper layer of the measured data sets, areas enlarging the thin conductive layer simulated in Fig. 9 are therefore interpreted as a signature of epikarst features.

Figure 10 summarises the interpretation of the spatial resistivity distribution made in the light of structural observations and ERT forward modelling. The 8 regions of Fig. 8 can now be named as follows: soil plateau (1),

epikarst plateau (2), epikarst doline (3), matrix plateau (4), porous limestone (5), massive limestone (6), clayey limestone (7) and fracture (8). Note that the highly conductive anomaly near the bottom of the doline is interpreted as the beginning of a fractured area dipping 30° to the North, as many of them are evidenced in the borehole logs.

### 7.2    Time-lapse ERT to image karst hydrological processes

In a limestone context such as the RCL site, low resistivities indicate either clay rich areas or porous areas in humid conditions. However, only the moisture is subject to change on an annual basis. With that in mind, the resistivity variations of the 8 regions defined in Fig. 8 can be tracked. The fact that the resistivity values of these regions seem to evolve distinctly regarding climate conditions points to different hydrodynamic behaviours coexisting very close to one another, in agreement with the percolating water discharge data sets. Figure 11

shows the temporal evolution of the median of the absolute resistivity values in the 8 regions. GD data sets were chosen for surface and subsurface regions (1, 2, 3 in Fig. 8) given the better resolution of that protocol at shallow depths. Resistivity values of the deeper regions (4, 5, 6, 7, 8) come from the DD surveys, because of the greater imaging resolution at depth and especially the inability of GD arrays to properly image the fracture anomaly. It results in fewer but better constrained data points. To compensate for the lack of DD surveys, especially during

winter 2016, corresponding GD data sets are added with a transparent mask as a guide in Fig. 11. Note that resistivity values of the surface layers from DD surveys and their temporal variations compare well with those of the GD surveys.

The first point to stress is that, as already visible in Fig. 7, all the regions experience a seasonal resistivity variation that seems related to the effective rainfall distribution. The time series displays two annual cycles:

April 2015 to March 2017. Data from 2014 comprise one survey in February, another one in March and a daily





series from April to June, which confirm the relatively low resistivity conditions attributed to the end of winter and the beginning of spring. This correlates well with the positive monthly effective rainfalls that are responsible for a general recharge of the soil moisture and groundwater reservoirs, hence resulting in lower resistivity values. A general increase in resistivity is noted when the effective rainfalls become negative in spring, with slight

delays in deeper regions. The slope of this increase varies by several orders of magnitude from one region to another; surface and subsurface regions showing the sharpest increases, especially on the plateau side of the ERT profile. The massive limestone region displays a seasonal variation with the highest absolute amplitude (note that Fig. 11 displays the log of resistivity), associated with the highest average resistivity. The area that shows the lowest seasonal amplitude is the clayey limestone region, with only 220 Ωm between the minimum in spring and

the maximum in winter. The greatest delay with respect to the seasonal dynamic of the surface layers is encountered in clayey limestone region. The porous limestone region on the plateau side exhibits an intermediate behaviour rather close to that of the massive limestone and the surface and subsurface slope of the doline regions, albeit of greater amplitude.

Lower seasonal amplitudes have two possible causes: higher minimum or lower maximum groundwater contents

reached in summer or winter respectively. In similar lithological compositions, if some regions experience lower groundwater deficits, this would result in lower resistivity variations. However, these regions exhibiting such low resistivity variations are also those characterised by low resistivity distribution (<500 Ωm). This calls for a second hypothesis, i.e. clay-enriched layers. According to Waxman and Smits (1968), the greater the clay content, the lower the bulk electrical resistivity amplitude between dry and saturated state. Observations in this

area reveal clayey limestone, which supports that second hypothesis.

### 7.2.1 Specifying limitations for relating resistivity variations to groundwater content changes

Converting resistivity variations to groundwater content changes is one particular advantage of ERT monitoring. However, this requires specific site characteristics in terms of subsurface homogeneity that a complex karst system may not offer. Figure 12 presents the co-evolution between data of the 10cm deep WCR and the

resistivity of the surface layer on the plateau side of the ERT profile. The porosity of the layer was estimated based on the maximum VWC value reached (0.33), which represents the wettest situation measured at a depth of 10cm. Assuming that this corresponds to a state very close to saturation, this provides a first estimate of the porosity in the top layer. Depending on the rainfall and PET conditions, the relationship between saturation and resistivity displays several distinct trends in Fig. 12. Each of the trend lines generally corresponds to a separate

drying episode that follows rainfall events, as highlighted by the time color map in Fig. 12. A particularly long trend line reaches very low saturation values. It corresponds to the August and September 2016 drought. Theoretical relationships derived from Archie's law (Archie, 1942) are also displayed, taking variable pore-water conductivities, as comparative standards. Other relationships requiring more parameters to be fixed (Garré et al., 2011; Revil et al., 1998; Rhoades et al., 1976; Waxman and Smits, 1968) have been tested, hence resulting in

similar conclusions: the existence of different resistivity/saturation trends highlights a variable pore-water conductivity. Especially, the August and September 2016 drought may be attributed to a progressive increase of the pore-water conductivity. Following a rainfall event, the part of the rain water that is not rapidly flushed in the matrix keeps mixing with pore-water while increasing ionic leaching. As a result, without significant rainfall events, groundwater resistivity progressively decreases together with an increasing presence of ions in solution.





The effect of the important aquatic subterranean fauna usually present in the epikarst layer (Pipan and Culver,
2005; Sket et al., 2004) on the electrical conductivity remains poorly studied. Given the low seasonal amplitude
of the electrical conductivity variations measured in the percolating water, it is however thought not to play a
major role at the RCL site.

As shown in Fig. 3d, the specific conductivity of the in-cave drip discharge shows a quite stable behaviour (~30
Ωm). Increases in pore-water resistivity after rainfall events however reach up to ~65 Ωm, which is attributed to
a greater income of fresh rain water directly percolating through fractures. The mixing between rainwater with
highly variable conductivity and groundwater may be either progressive through the vadose zone or concentrated
in the epikarst. In any case, the significantly lower conductivity of rainwater and the increase in groundwater
saturation after rainfall events have opposite impacts on the measured apparent resistivity of the rock. Such a
process may therefore tend to underestimate moisture contents deduced from electrical resistivity data after
rainfalls. It is responsible at least for a greater uncertainty that enlarges the error on estimated moisture content
variation.

Therefore, Fig. 12 illustrates the complexity inherent to the estimation of water content from ERT data even if
interesting interpretations may come out of case studies that assume constant pore-water conductivity (e.g. Beff
et al., 2013; Brunet et al., 2010; Chambers et al., 2014; Garré et al., 2011; Michot et al., 2003). Such assumption
can clearly not be the stated here. Likewise, Uhlemann et al. (2016b) point out similar limitations, attributing
abnormal resistivity changes in wetlands to pore-water conductivity variations, which also results in the
inapplicability of converting bulk electrical resistivity to moisture contents.

In such a heterogeneous karst context as the RCL site, other important limitations for estimating groundwater
contents derived from ERT data concern the porosity, the clay content and the calibration of fitting parameters of
petrophysical relationships (Archie, 1942; Revil et al., 1998; Rhoades et al., 1976; Waxman and Smits, 1968).
Additionally, absolute electrical resistivity values imaged after inversion, especially for high resistivities, remain
dependent on inversion parameters and resistivity contrasts, which mitigate the accuracy of the results. In the
absence of precise calibration factors, determining groundwater contents from ERT measurements, even time-
lapse, remains highly challenging in a karst environment. Nonetheless, the identification of variable dynamics
attributed to groundwater content changes in different spatially limited areas of the subsurface may help in
developing hydrological models applied to the vadose zone of complex karst systems.

### 7.2.2    Seasonal recharge processes

To be able to qualitatively analyse the temporal resistivity behaviour of each defined region as a characteristic of
their hydrological dynamic, Fig. 13c and d presents the median of difference in log resistivities with respect to
dry conditions (15[th] October 2016), as defined in Equation (1), for the plateau side and the doline side regions
respectively. This allows effective comparison of the resistivity evolution from each region and with
environmental data. As pointed out by Samouëlian et al. (2005), looking at changes in resistivity rather than the
absolute values eliminates systematic errors. Data from three of the five WCRs are shown in Fig. 13a. Data of
the three drip discharge stations are presented in Fig. 13b while positive effective rainfalls are displayed in Fig.
13e.

Effective rainfall and data of the 10cm WCR probe are affected by the surface weather conditions as mentioned
in Sect. 3. At first sight, three resistivity dynamics can be identified with respect to surface conditions: (D1)





regions that correlate the most with the shallow WCR probes, and thus with the fracture drip discharge (PWD1), (D2) regions that exhibit a delay in the increase and the decrease of resistivity, hence correlating more with the deeper WCR probes and especially with the matrix (PWD2) and stalactite (PWD3) drip discharge, (D3) the fractured region that, despite showing a damped behaviour close to the second group, is characterised by higher variability in response to separate rainfall events.

Surface and subsurface regions on the plateau side of the profile, attributed to the soil and epikarst, are very dynamic, hence are members of the D1 type. This is in accordance with Klimchouk (2004) that defines the epikarst as a dynamic system. The sharp decrease following rainfall and progressive increases during every period without significant rain testifies to the strong hydrological relationship with the atmosphere, i.e. precipitation and evapotranspiration. The soil and epikarst layers have their moisture contents directly monitored with the WCR probes. A decrease in resistivity is noticed following rainfall events, usually for effective rainfall greater than 2 mm. Such a behaviour is typically observed in soil layers and has been reported in numerous ERT monitoring studies, including karst landscapes (e.g. Brunet et al., 2010; Carriere et al., 2015). Michot et al. (2003) specifically focused on the estimation of soil water content using time-lapse ERT measurements. Seasonally, after large increase in resistivity through the summer period, recharge of these upper layers is observed and starts in September 2015 and October 2016. A minimum resistivity is reached in December 2015 and March 2017. This seasonal offset is due to the particularly dry fall 2016 and winter 2017. Minimum resistivities should be associated with maximum amounts of groundwater content.

The relationship between the WCR data and the fracture drip discharge (PWD1), already highlighted in Fig. 3, may therefore be extended to the entire soil layer of the RCL site. The subsurface layer must be characterised by a lower porosity and clay content given its higher average resistivity as evidenced by Fig. 11. It shows however a dynamic very comparable to that of the surface layer. This highlights the great hydraulic connectivity between the soil layer and the epikarst, and hence typically seen as the principal reservoir that feed vadose flows (Arbel et al., 2010), in this case the fracture drip discharge station (PWD1). In the presence of a thin soil layer, as it is the case at the RCL site, underlying materials, e.g. the epikarst, must act as a reservoir for a considerable amount of groundwater available for the vegetation (Williams, 2008). However, the fact that the average resistivity progressively increases from the surface to the subsurface layer must also be interpreted in the light of the smoothness constraint of the inversion.

The surface of the slope of the doline also displays a D1 type dynamic, albeit more damped. Sharp decreases in resistivity are still observed following major rainfall events, but the seasonal dynamic is less marked, with a less clear correlation with the WCR probes. This is most likely due to (i) runoff processes occurring in this zone given the strong topography, or (ii) to a very poor capacity in terms of water retention because of the progressively thinner soil layer. The fact that no specific resistivity variations are noticed in Fig. 7, next to the first five electrodes at the bottom of the doline, is in agreement with the absence of soil layer in that area and the subvertical topography where surface runoff is likely to be predominant. This justified why this area was not included in the surface region tracked in the slope of the doline as displayed in Fig. 8, Fig. 11 and Fig. 13.

D2 dynamics are attributed to the deeper regions comprising the clayey limestone, porous limestone, massive limestone and the matrix to the North of the plateau. The three latter regions exhibit a very similar seasonal variability; the massive and porous limestone regions reaching their maximum approximately at the same time, ~25 days after the D1 types (Fig. 13c-d). The matrix on the plateau side shows a slight negative offset, reaching



its maximum closer to that of the D1 types. These D2 type regions however stay in high resistivity conditions for approximately 75 more days: until end November 2015 and end January 2017, following a particularly dry fall/winter. Conversely, the clayey limestone, despite a much lower seasonal amplitude, reaches its maximum resistivity more than 4 months after the D1 types. Overall, the similarity between the D2 curves and PWD2 or PWD3 data sets is indubitable. As shown in Fig. 13, the top of the D2 curves correspond to dry periods identified from drip discharge data, except for the clayey limestone for which it corresponds only to resistivity increases. The reactivation of groundwater recharge in this layer, evidenced by a general decrease in resistivity in winter can be correlated to the reactivation of PWD2 and PWD3 high drip discharge regime. For the other D2 type regions, the curve starts decreasing when the slope of PWD3 curve turns positive in dry periods, i.e. in December 2015 and January 2017 (Fig. 13c-d).

A delay is also noticeable at the beginning of the resistivity increase, especially during the 2016 dry period, in the porous limestone, and more strongly in the massive and clayey limestone curves. This lag period reaches 15 days in August 2016. Such trends attest that these deeper regions stay less affected by the surface conditions in the beginning of drying processes. This indicates of a delayed infiltration in deeper areas. Then the rise in resistivity is significant only after the upper layers reach ~75% of their increases; the surface of the plateau side having already risen from -20% to -5% change in log resistivity. Similarly, the decrease of the D2 curves takes place mostly during winter, when the upper layers are already lowered close to their minimum value.

The fractured region also exhibits a dynamic close to the D2 type, yet on which a greater variability in response to rainfall events is superimposed. Such resistivity decreases are however different from the response of surface layers, that typically shows a sharp decrease depending on rainfall intensity followed by a slower increase, which corresponds to soil drying processes evidenced by the WCR data. In the fractured region, the resistivity perturbation induced by rainfalls is more ephemeral; the resistivity curves retrieving most of the time the resistivity value prior to precipitation 1 day after the rainfall event, which mimics the sharp recession curves following rainfall events displayed by PWD1 data.

### 7.2.3    Resistivity response to rainfall events

Figure 14 particularly focuses on ERT data at the rainfall event scale, showing cross-correlation functions of the eight resistivity regions with the environmental variables on a lag window of -5 to +5 days. In order to correlate consistently these different types of data sets, the first derivative of the resistivity time series was selected and cross-correlated with the rainfall, the first derivative of the 10 and 105 cm WCR data, and the first derivative of the log PWD data sets. The first derivative highlights the variations of the measured variables from one day to the next, while centring the time series on 0. Taking the log of the PWD data sets provides a way to account for minor variations in dry periods. To take into account the different sampling rates of all the data sets, cumulative rainfalls and mean WCR and PWD values are computed within 8-hour windows from 6 hours prior to 2 hours after each ERT time step. Such time windows account for the average delay necessary for a rainfall event to be detected in the 10 cm WCR probe and PWD1 data, which are the monitored spots that are most reactive to infiltration of rain water.

As a result, most of the correlation peaks occur at lag 0 or lag 1 day. No significant peaks are noticed at negative lags, except for the clayey limestone region that exhibits a behaviour totally different from the rest. This will be discussed further in this section. For other regions, no interpretation can therefore be drawn concerning





variations of resistivity that could precede variations of the environmental variables. D1 type regions exhibit very similar patterns, being unsurprisingly most negatively correlated either to the rainfall, the 10cm WCR or the PWD1 data sets at lag 0 day (Fig. 14a-b-d). This means that decreases in resistivity are mostly concurrent, with respect to the temporal resolution of 1 day, to rainfall events as well as increases in soil moisture content or drip discharge (see Fig. 13). Logically, the cross-correlation function rapidly decreases with positive lags for the correlation with rainfall and the 10cm deep WCR data sets. Resistivity data of the D1 type regions also exhibit non-lagged cross-correlation peaks, although smaller, with the 105cm WCR and PWD2 data (Fig. 14c-e). No correlation comes out for any of the regions with PWD3 data (Fig. 14f), which highlights its non-reactivity to storms during dry periods, as visible in Fig. 3. Interestingly, the fractured region exhibits cross-correlation functions similar to those of D1 types region, albeit more damped, displaying peaks with negative cross-correlation coefficients at lag 0 day with all the variables except the PWD3 data. However, the fractured region becomes positively correlated as early as lag 1 or 2 day, suggesting a rapid resistivity increase following a first decrease concurrent to a rainfall event or VWC increases. That particular trend of the fractured region is also visible for the cross-correlation with the PWD1 and PWD2 data sets.

In contrast to this, D2 type regions exhibit patterns different from each other and from those of D1 types regions. The matrix plateau and porous limestone region evolve quite similarly with a cross-correlation peak at 1 day lag for the rainfall, 10cm WCR probe and PWD1 data sets; the matrix plateau region still showing a noticeable cross-correlation coefficient at lag 0 day. This particular region is characterised by a succession of thin massive and porous limestone. The greater number of bedding planes could therefore provide more preferential pathways favouring rapid percolation of rainwater, compared to the porous limestone region. Both regions are however correlated without any lag, with the deepest WCR probe and PWD2 data at lag 0, similarly with D1 types regions; the matrix plateau region being even the most correlated with these variables. This confirms at short-time scale the greater contribution of these D2 type regions to the PWD2 seasonal drip discharge.

The porous limestone and rainfall cross-correlation coefficient decreases progressively, being still negatively correlated at positive lag 2 and 3 day. The resistivity of the porous limestone still decreasing 2 to 3 days after a rainfall event is a possible explanation, highlighting a likely diffusion of the rainwater infiltration within the porous limestone layer. Despite being associated with the D2 type regions on a seasonal basis, the massive limestone does not show any clear correlation with any of the environmental data at the rainfall event scale. This is likely due to the very low to zero rainwater contribution to the bulk resistivity of that area given its low porosity and fracture density.

Finally, the clayey limestone shows a quite surprising trend, being positively correlated at lag 0 with most of the environmental data. In other words, increases in resistivity are concurrent with rainfall events. This is also particularly visible in Fig. 7b-5 and b-10. We propose two possible explanations of this observation: (i) a time-lapse ERT inversion artefact; (ii) an influx of rainwater more resistive than the actual pore-water.

(i) The first hypothesis pointing out time-lapse ERT inversion artefacts was already addressed in the case of shallow infiltration monitoring by Clément et al. (2009). They demonstrated using synthetic models that infiltration processes at shallow depths usually produce a decrease in resistivity in the upper layer, while an increase in resistivity may be observed at intermediate depths, where the resistivity is actually not changing. Descloitres et al. (2008) identified such time-lapse artefacts, when tracking seasonal recharge processes by ERT, especially located in the subsurface of slopes. These studies point out that infiltration through a succession of



layers with different resistivity signatures may enhance such artefacts. Hence, the relatively complex resistivity distribution in the subsurface of the slope of the doline may favour such artefacts, as the conductive clayey
limestone lies below a more resistive layer, which is itself below the surface conductive layer. According to Clément et al. (2009), a proposed solution to avoid such artefacts involves adding decoupling constraints in the inversion along a shallow line evolving together with the infiltration front, determined by external data. This seems however in our case quite unrealistic given that the evolution of the infiltration front remains unknown in the clayey limestone. The plateau is likely to provide a major part of the water infiltrating the clayey limestone,
given the strong run-off processes expected in the slope of the doline. Hence, a minor part of infiltrating water is still expected to come from the sinkhole itself, especially through open fractures. The hydraulic conductivity is also constrained by the lithological nature of each layer, which complicates the problem. Therefore, the complex infiltration in the clayey limestone justifies not adding such a decoupling line in the inversion. Alternatively, artefacts could also result from the underestimation of the noise (LaBrecque et al., 1996b) when weighting the
observed data of each data set for the inversion. The use of reciprocal errors for the DD surveys (used for tracing changes in the clayey limestone in Fig. 11, Fig. 12 and Fig. 13) must however mitigate this possibility.

(ii) The latter proposed hypothesis regarding these increases of resistivity following rainfall events in clayey limestone must also be discussed. In some cases, an influx of a great quantity of fresh, hence less conductive, water into a partially wet clayey material can result in an increase in resistivity, especially given the inverse
power relationship between the bulk electrical resistivity and the saturation (see Fig. 12), as evidenced in the model proposed by Waxman and Smits (1968). The only option being that the clayey limestone stays constantly at a high saturation rate, with highly conductive pore-water (~30 Ωm as shown by the drip discharge electrical conductivity data). An influx of more resistive percolating rainwater (> 100 Ωm), could therefore result in a slight increase in resistivity, even if the saturation rate also increases. While this could occasionaly happen
following precipitation with particularly low conductivity rainwater, this is however unlikely to be predominant in the signal.

### 7.3    Resistivity dynamics as markers for karst hydrology

Overall, the observations concerning the resistivity dynamics of each region are determinant, as they provide an image of the sources of the drip discharge type measured in the cave system. Especially, the nature of each
region and its variability in terms of resistivity must be interpreted in conjunction with its likely contribution to the karst hydrodynamic at the RCL site. Firstly, the role of the soil and epikarst is evidenced as being very dynamic (D1) with regard to the atmospheric inputs. This reactivity indicates a limited buffering role for the epikarst in terms of hydrological recharge, as evidenced by previous studies (Aquilina et al., 2003; Bakalowicz, 2005; Ford and Williams, 2008; Genty and Deflandre, 1998; Poulain et al., 2015; Sheffer et al., 2011). Indeed,
the likely very thin epikarst in the middle of the plateau as pointed out by the resistivity images and other field observations strongly limits water storage. Based on the resistivity variations, the epikarst, thicker to the North of the plateau, acts anyway as a buffer during spring, being still a likely source of diffuse infiltration to deeper areas while responding to the water demands of the vegetation. Storms cause ephemeral recharge in these surface layers but also increase rapid infiltration to deeper areas as revealed by the dynamics of the fractured area
following rainfall events, even in summer. This second effect seems however damped during the summer.





Diffuse flows propagating through the rock matrix are also evidenced with D2 dynamics, which is consistent with previous findings (e.g. Lange et al., 2010; Perrin et al., 2003). Carrière et al. (2016) particularly highlighted via ERT monitoring of rainfall events that considerable amounts of water can be temporarily stored in the vadose zone of karst systems, and more specifically in the porous matrix. At the RCL site, the fact that all regions reach

a minimum threshold in winter periods is also interesting, meaning that they approach their maximum water retention capacity. The relatively weaker effects of winter rainfalls on the resistivity variations compared to those of summer storms testifies to the high saturation present in the rock matrix during such periods. Given the inverse power relationship between saturation and bulk electrical resistivity (Archie, 1942; Revil et al., 1998; Rhoades et al., 1976; Waxman and Smits, 1968) as illustrated by Fig. 12, progressive increases in saturation are

expected to result in lower and lower resistivity decreases, as for the drying period of August and September 2016. This means that in relatively porous rock, such as the porous limestone at the RCL site, an important recharge occurs in winter. Due to the aforementioned limitations in accurately estimating groundwater contents, the intensity of the deficit, when the maximum of resistivity changes is reached, remains unconstrained.

The role of the conduit porosity is also clearly evidenced by the greater variability in the fractured region,

compared to that of the massive limestone and porous limestone. D3 dynamics can be interpreted as an evidence of the direct hydraulic connectivity between surface layers and post-storm drip discharge of the percolating water (Arbel et al., 2010) as monitored in-cave. Furthermore, the seasonal variation revealed in the fractured zone is an indicator of the likely water exchange from the conduits to the matrix porosity in that area. Enhanced porosity near fracture walls as seen in core samples can temporarily increase stored groundwater. When the fractures are

saturated with percolating water after rainfall events, a gradient towards these porous areas may occur, as already modelled in previous studies (Bailly-Comte et al., 2010; Hartmann et al., 2013). Similarly, an input of water from porous areas close to the fracture walls or from crosscut porous layers is likely to occur in dry periods, participating to the vadose baseflow observed in PWD1, as Poulain et al. (2017) found at the RCL site. Alternatively, a slower infiltration can also occur in narrower fractures or in bottlenecks towards the main

fractures.

Figure 15 summarises these interpretations and the hydrological functions of each of the regions imaged by the ERT monitoring at RCL site.

## 8    Conclusions

This first long-term permanent ERT monitoring of a complex karst vadose zone has revealed seasonal recharge

processes with variable dynamics. ERT allowed clustering of distinct areas showing contrasting evolutions through three hydrological cycles. Such different behaviours are attributed to distinct categories of vadose reservoirs responsible for specific percolation types. They could be associated with the sources of distinct percolation drip discharge measured in-cave. This study was able to differentiate 3 groups with distinct resistivity variations and to link them to sources of in-cave drip discharge:

• Upper conductive layers, comprising the soil and the epikarst, which are in direct contact with the atmosphere, hence showing the highest variability (D1).

• Deeper regions characterised by a more diffuse and damped seasonal variation, showing a delayed recharge (D2). The resistivity values of these areas depend on the lithology, the porosity and the clay content, which also determine their groundwater retention capacity.





• A conductive fractured zone (D3) exhibiting a dynamic close to that of D2 group, but with a greater variability in response to rainfall events, revealing a preferential pathway for rainwater. Water exchange between conduits and porous matrix is likely to explain the seasonal variation.

These observations are consistent with previous knowledge of hydrological processes in karst vadose zone (Ford and Williams, 2008; Goldscheider and Drew, 2007), while bringing a first detailed view of the sources

responsible for the duality of flows typically observed in karst environments: quickflows and diffuse flows (White, 2002). Moreover, given the small resistivity variations measured in winter, the recharge processes in all areas of the monitored site are expected to be highly efficient. The main constraint on the amount of groundwater volumes stored in the vadose zone appears to be the matrix porosity.

Hydrogeophysical experiments in karst systems, especially targeting the vadose zone, are very challenging, as

already raised by Chalikakis et al. (2011). This study proves that, combined with a detailed structural and lithological survey at a local scale, as well as with additional environmental measurements, ERT monitoring is able to image and track through time recharge processes within the vadose zone of a karst system.

Improved ERT inversion strategies for highly heterogeneous environments could provide more constrained results reducing the occurrence of artefacts similar to those experienced in the clayey limestone of our field site.

Synthetic modelling approaches could also help validating the assumptions made on the water infiltration processes. In parallel, 3D imaging would improve spatial resolution, and hence, reducing uncertainties as compared to 2D imaging. In such case, a real effort should however be made for optimizing more complex measuring protocols with regards to highly resistive karst environments.

Overall, these findings support existing models and bring new opportunities to understand the karst system, often

modelled as black boxes. Imaging the sources of drip discharge signals conventionally monitored in numerous cave systems contributes to improve the understanding of karst subsurface hydrodynamic behaviours. More specifically, this study calls for similar geophysical monitoring to be tested in different karst environments, in combination with conventional hydrological monitoring or dense monitoring networks of cave drip water (Mahmud et al., 2016), to gather new types of hydrological data to be included in karst hydrological modelling.

Such novel approaches are required to face future challenges for the management of karst groundwater resources worldwide and the increasing risks of contamination issues raised by the increasing agricultural demands. Better constraining recharge processes of karst aquifers also brings grist to the mill of the study of speleothems, with implications on paleoclimatic researches.

## 9    Data Availability

Data used in this research paper (comprising ERT, cave drip discharge, soil moisture, rainfall and water conductivity) are available upon request to the first author (arnaud.watlet@umons.ac.be). They will be deposited on a public data repository following the acceptance of this manuscript. ET$_0$ data are available via Pameseb organisation (pameseb.be).

## 10    Acknowledgements

This work is part of the Karst Aquifer Research by Geophysics (KARAG) project (karag.be) funded by the Fonds de la Recherche Scientifique (FNRS-FRS). Arnaud Watlet is also supported by the Ernest du Bois Fund,



managed by the King Baudouin Foundation. We would like to thank all the colleagues who were involved in this project and most particularly Claudio Barcella for field and laboratory works, Christophe Bastin, Bérénice Deletter, Fadel Cissé, Stefaan Castelein, Marc Hendrickx, Giovanni Rapagnani, Sebastian Uhlemann, Fabian Wagner, Thomas Lecocq, Koen Van Noten and Thierry Camelbeeck, as well as the ASBL Grottes de Lorette, Guy Evrard and the team members, and the municipality of Rochefort for their hospitality and support. This paper benefited from Matplotlib (Hunter, 2007), Pandas (McKinney, 2011), Glue (available at glueviz.org) and GDAL/OGR packages (available at gdal.osgeo.org) as well as QGis (available at qgis.osgeo.org) software.


## 11 Annexes

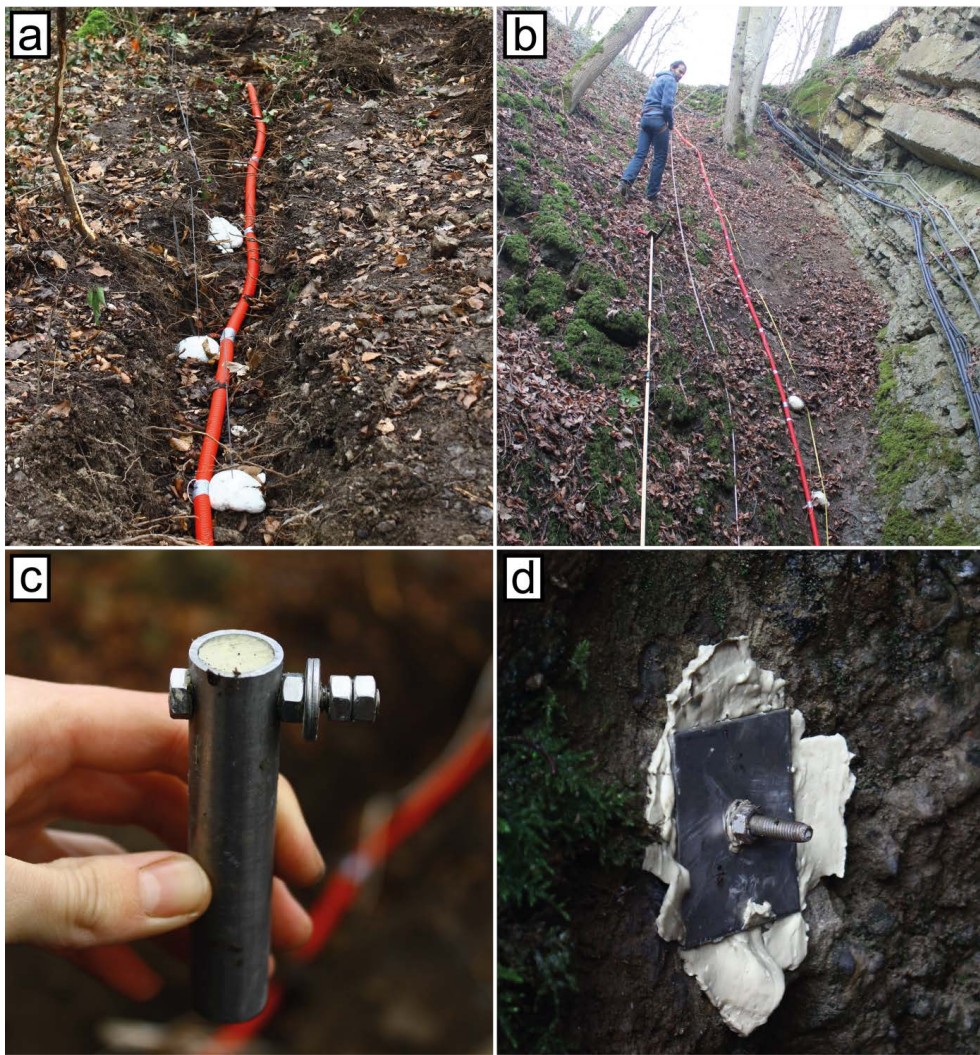


**Figure A 1 Photos of the ERT profile, with the trench at the top of the plateau (a), the situation in the slope of the doline (b), the stainless steel hollow tubes used as electrodes in most of the profile (c) and the stainless steel plates bolded in the rock at the bottom of the doline (d).**

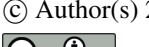



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





**Figures**

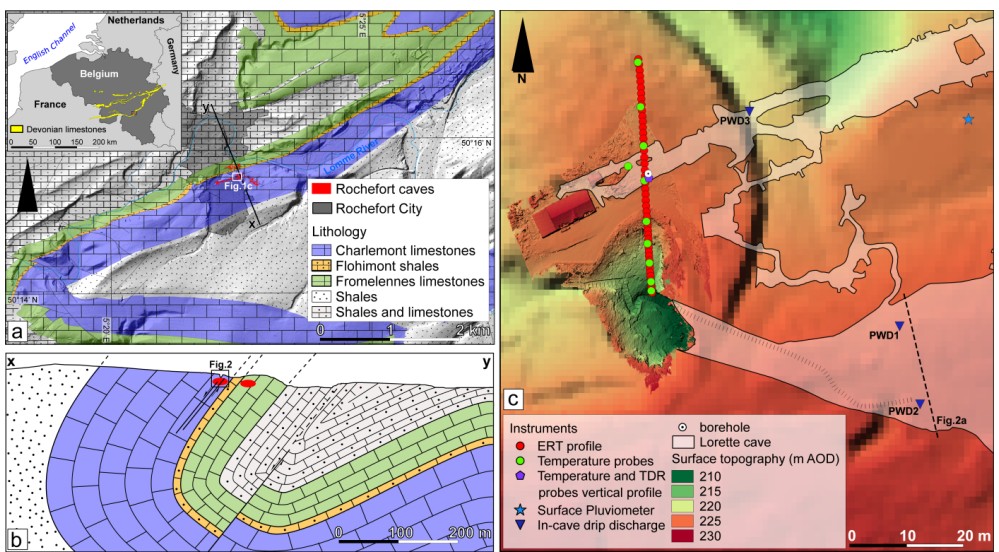

**Figure 1: a. Simplified geological map of the Rochefort area, highlighting the limestone formations and the Rochefort caves. b. Geological cross-section (x to y in 1A), modified after Delvaux de Fenffe (1987). It displays the overturned syncline marked by high dipping sedimentary layers (N070-S50) in the Rochefort caves area as well as active normal faults evidenced by Vandycke and Quinif (2001). c. General overview of the Rochefort monitoring site. Electrodes from the ERT profile, WCR and temperature probes as well as the rain gauge are shown. Background 1 m Digital**

**elevation model (based on Lidar data of DGO3-SPW) are replaced by a high resolution surface topography model from a 3D photoscan (Triantafyllou et al., 2017). This highlights the monitored sinkhole that gives access to the Lorette Cave (in white).**





**Figure 2: a. Cross-section of the *Val d'Enfer* room (location highlighted as a dashed line in Figure 1c), clayey layers (black) and porous limestone layers (dark grey) are highlighted in the massive limestone pile (light grey). Fracture with evidence of displacement are noted in plain red. b. Stereogram (Schmidt lower hemisphere; left) and rose diagram (right) of the planes attributed to S0 (blue) and fractures (red), deduced from statistical analysis of a 3D model of the *Val d'Enfer* room. C. Cross-section corresponding to the ERT profile (see location in Figure 1) based on logging of core samples and image survey of the borehole. Main plans are highlighted in green (S0 discontinuities or parallel fractures). Dashed green represent the extrapolation of two main fractures towards the surface of the profile. D. stereogram (Schmidt lower hemisphere; bottom) and rose diagram (up) of the plans highlighted in green in 2c.**



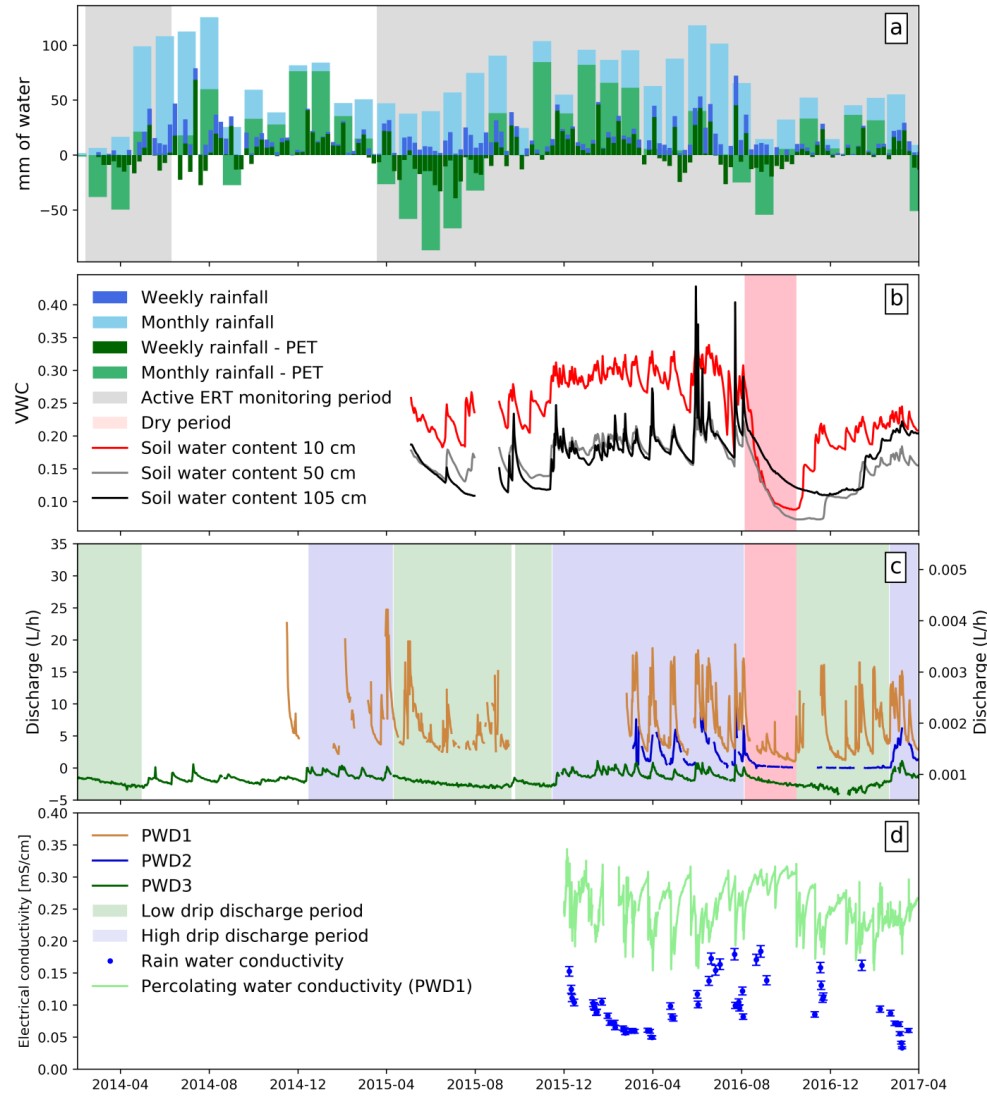

**Figure 3: a.** Weekly and monthly rainfall as well as effective rainfall (estimated as rainfall – PET). Shaded areas represent the periods with active ERT monitoring. **b.** Three of the five WCR probes data set for the 10, 50 and 105cm depths. The long drought of August and September 2016 is highlighted in pale red. **c.** Dataset for the three percolation water discharge (PWD) stations. Periods of low discharge and high discharge are highlighted based on values from the PWD3 and PWD2 when available. **d.** Electrical conductivity of rain water (for rainfall event > 5mm) and percolation water measured at PWD1 station. Error bars stand for the instrumental error announced by the manufacturer (±5%).




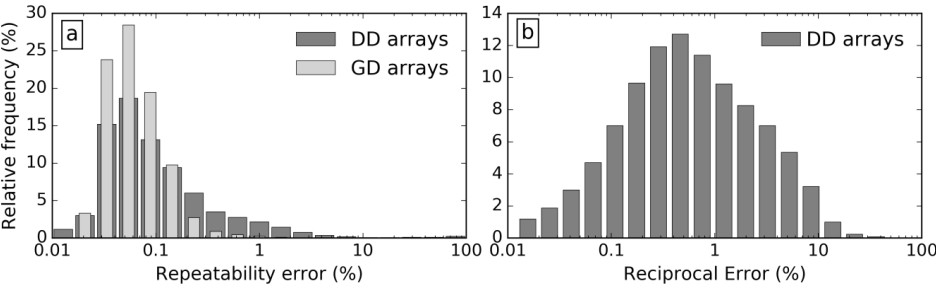


**Figure 4: Relative frequency of (a) the repeatability (stacking) error for the whole GD and DD datasets, and (b) the reciprocal error for the whole DD dataset.**

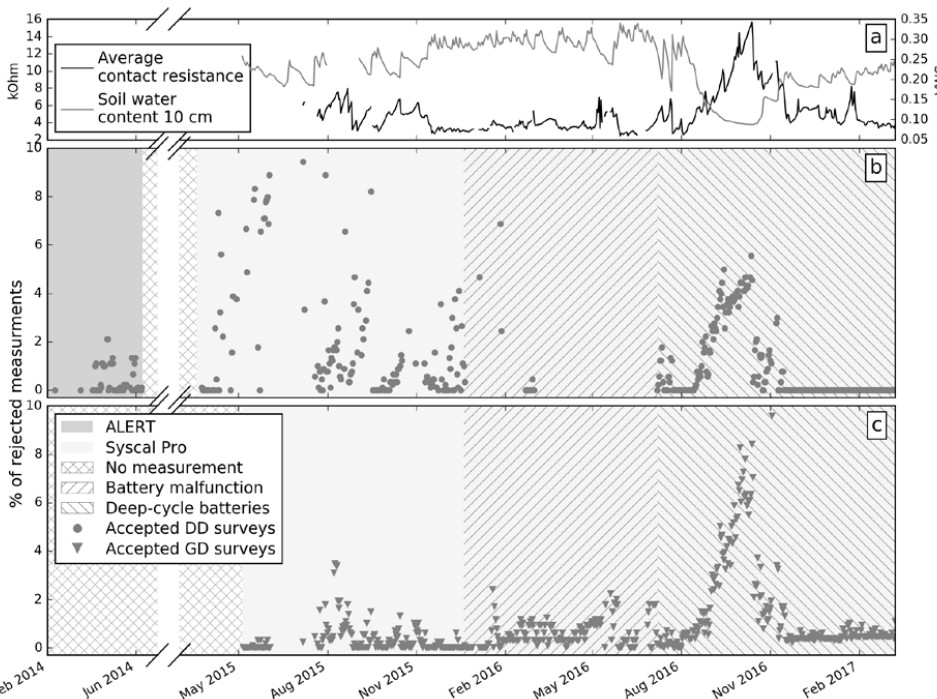



**Figure 5: A. Evolution of the contact resistances averaged along the ERT profile and the soil water content of the 10cm probe (see point 2.6). B. DD surveys used for the study, after filtering of the available datasets. Each dataset with more than 10% of rejected data (less than 810 accepted reciprocal measurements) were removed. C. GD surveys used for the study, after filtering of the available datasets. Each dataset with more than 10% of data rejected (less than 1180 accepted measurements) was removed. Note the break in the x axis between June 2014 and March 2015 where no measurements were acquired. Malfunction of the battery recharge made most of the DD available datasets to be rejected between mid-December 2015 and end of June 2016. The drying period in August and September 2016 also lead to a greater number of rejected data. The lower maximum measurement per survey after August 2016 is due to bad contact of electrode #12.**





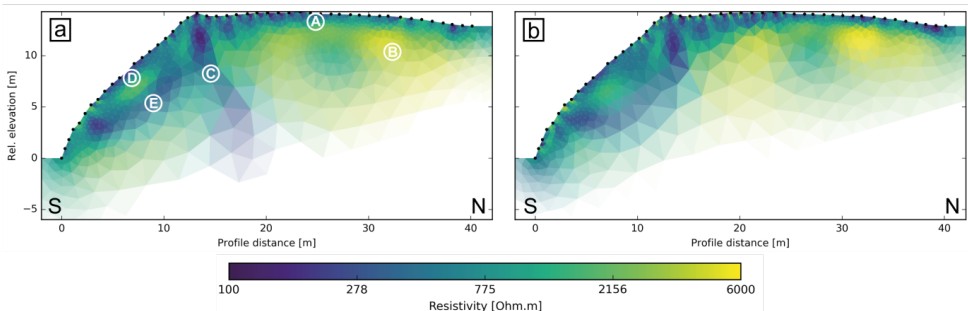

**Figure 6: Image of the inverted resistivities for DD (a) and GD (b) data set in average weather conditions (March 2017).**




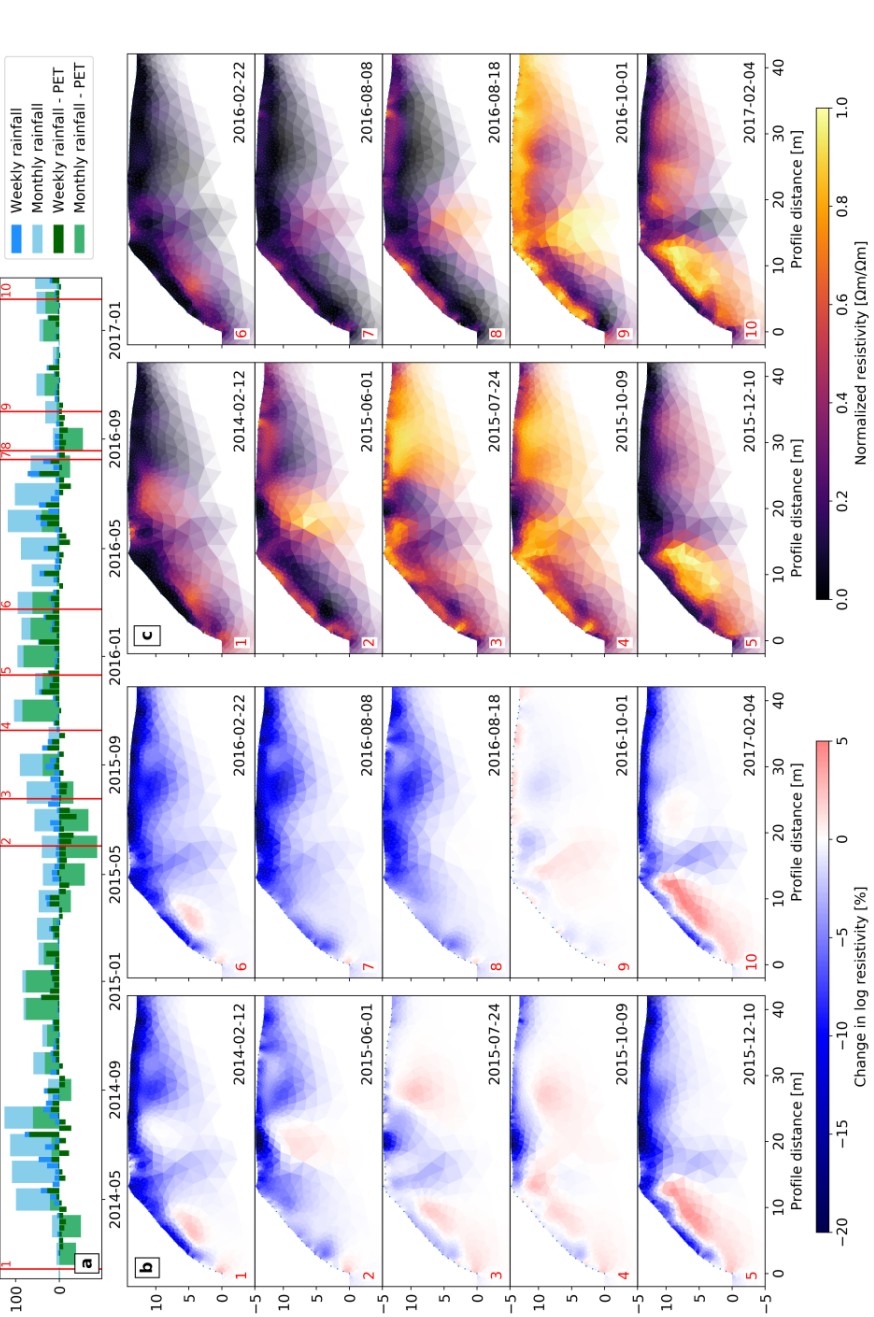

**Figure 7: a. Rainfall and effective rainfall. b. Changes in log resistivity (see Equation 1) for 10 of the 467 DD data sets, as highlighted in Fig. 7a. c. Normalised resistivity (see Equation 2) for the same data sets.**




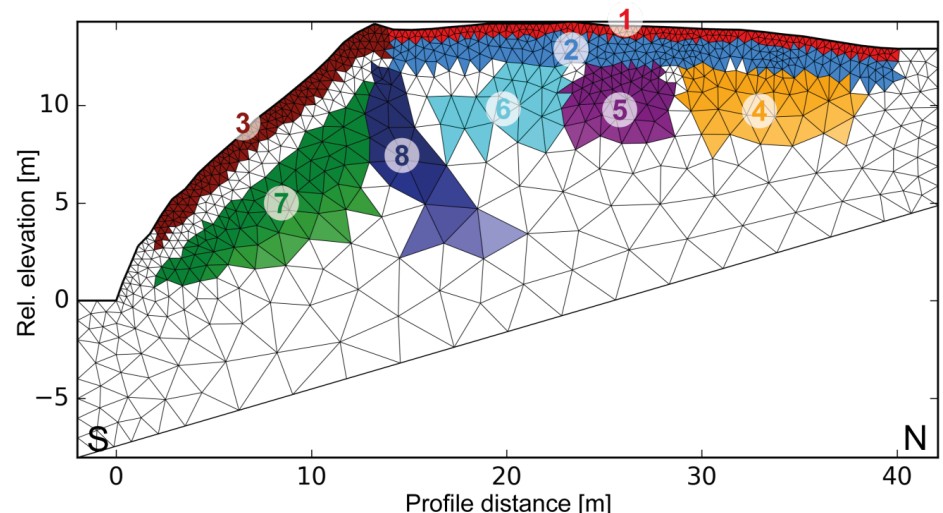

**Figure 8: Subdivision of the ERT image in 8 distinct regions based on their average resistivity values and arbitrary thresholds.**

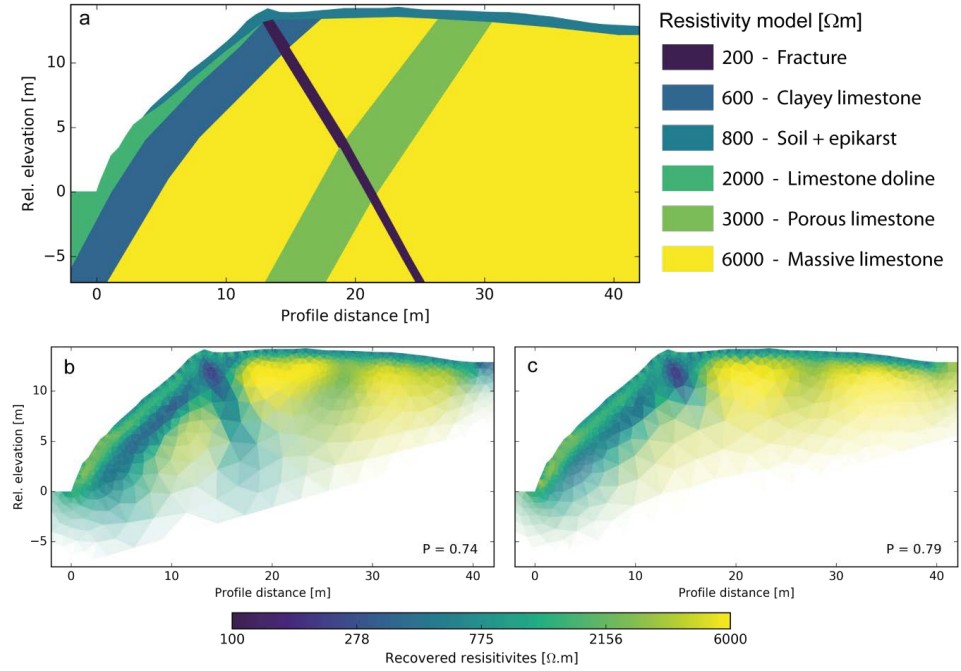


**Figure 9: Results of the forward modelling of the resistivity model (a) for the DD protocol (b) and the GD protocol (c). Pearson's correlation coefficients (*P*) are computed for the datasets of March 2017 presented in Figure 6.**





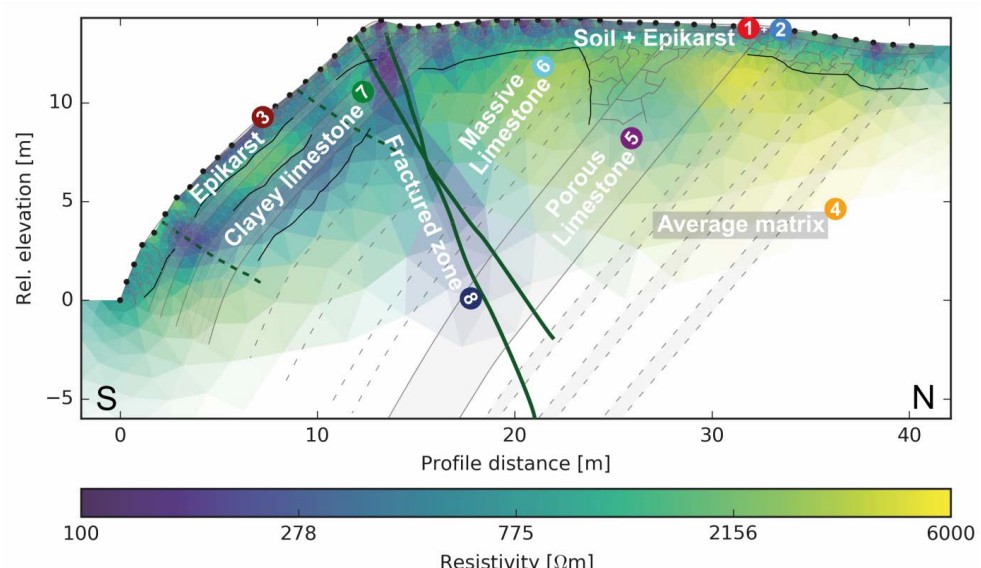

**Figure 10: Superposition of the geological model from Figure 2 on ERT results from March 2017, highlighting the interpretation of the resistivity distribution and anomalies.**

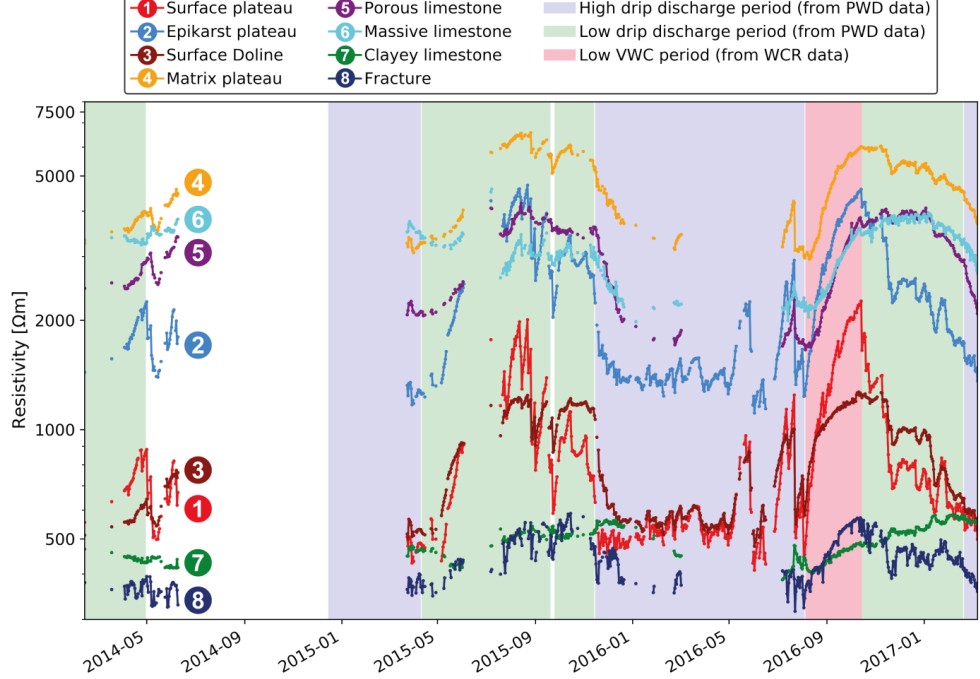

**Figure 11: Resistivity time series for the 8 regions detailed in Figure 10, expressed as the median of the log resistivity in each region. The median was chosen, rather than the mean, to limit the contribution of extreme values not representative of the robust central tendency of the cells of each region, especially at their boundaries. The dry and wet periods described in Figure 3 are also included.**



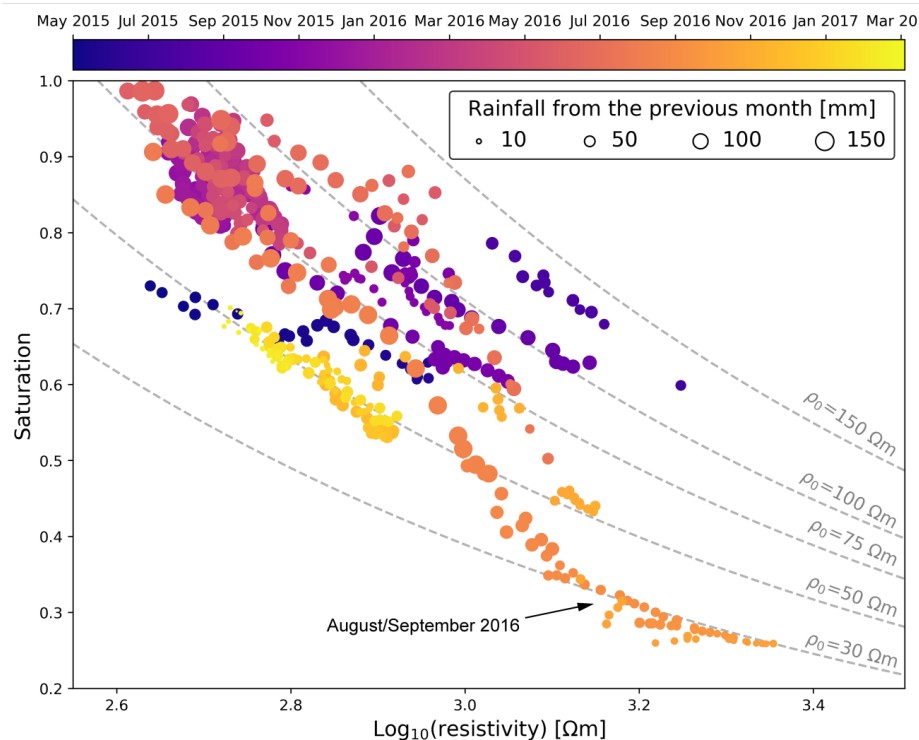

**Figure 12: Relationship between the resistivity (median) of the surface limestone region and the saturation from the 10cm deep WCR data. Theoretical relationships derived from Archie's law with successive pore water conductivity values are displayed as watermarks (Archie's cementation and saturation parameters are set to 1.5 and 2 respectively).**




**Figure 13: a. VWC data sets for the 10, 50 and 105cm deep WCR probes. b. Percolating water discharge of the PWD1, PWD2 (left y axis) and PWD3 (right y axis) data sets. c and d. Evolution of changes in log resistivity relative to dry period (15th October 2016) for the 8 regions from Figure 10. d. Daily effective rainfall. Daily rainfall greater than 3 mm are also included as spans in the whole figure (transparency as a marker of rainfall intensity) as well as dry and wet periods from Figure 3.**





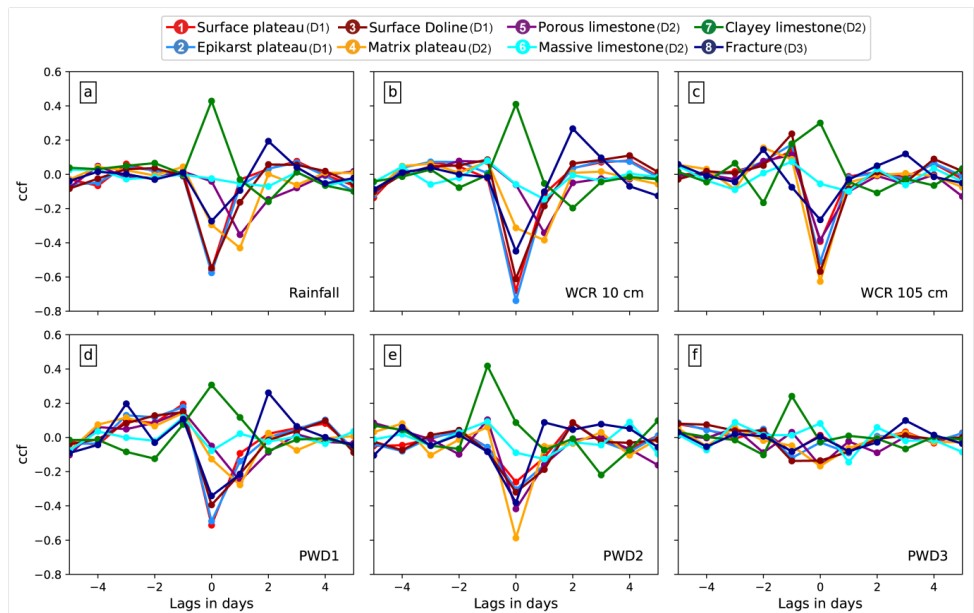

**Figure 14: Cross-correlation of the ERT data sets of the 8 regions and environmental data sets.**

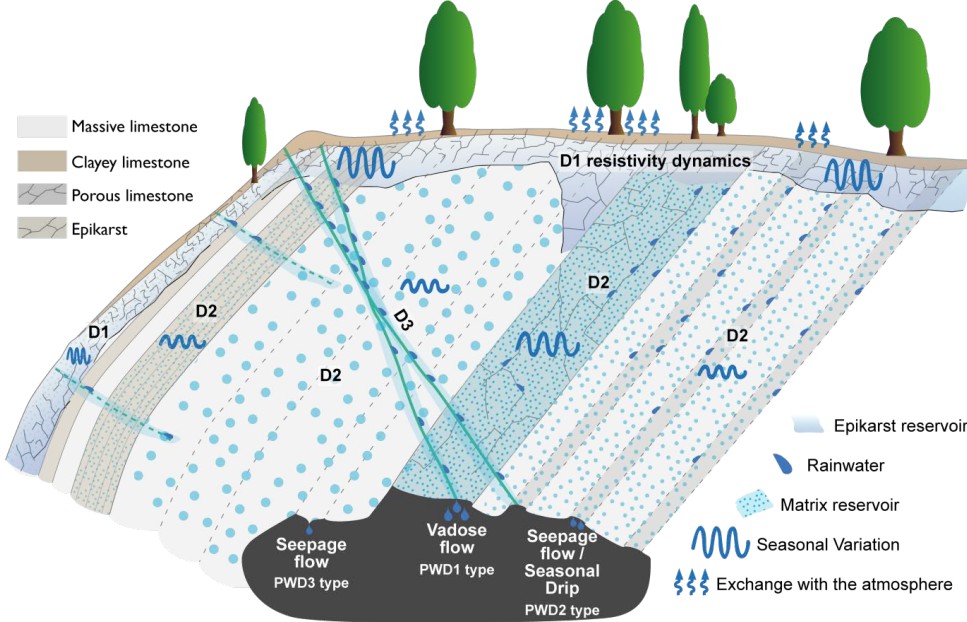

**Figure 15: Schematic view of the hydrological processes occurring in the subsurface of RCL site.**