# Peer review of "Imaging groundwater infiltration dynamics in karst vadose zone with long-term ERT monitoring"

_Hydrology and Earth System Sciences, 2017_

## Referee Comment (RC1) · Anonymous Referee #1 · 19 Sep 2017

**General comments**

This review will focus on the structure, context, and implications of the paper with respect to karst hydrogeology, rather than on specific comments on ERT methods and analysis, as I am not well-acquainted with ERT techniques. This paper presents and makes freely available valuable data from long-term, high-resolution geophysical monitoring of groundwater flow patterns in a karst system. These data support well-accepted conceptual models for infiltration and recharge processes in karst systems, particularly with respect to the role of the epikarst. The authors do not put forward new conceptual models, but present the data and their analyses and conclusions in high-quality, detailed, well-developed figures, which are a major highlight of the paper. However, the overall quality of the writing and the organization of the paper lack clarity

and concision. The paper would benefit from thorough, streamlining-focused editing, and from clearer explanations of the underlying concepts and assumptions the study is based on. Additionally, the authors propose one specific interpretation of the ERT data, but their analysis might benefit from considering what other interpretations would be possible, and what hydrogeologic data would be needed to test competing interpretations against each other and against additional data. Finally, although the dataset being presented is extremely impressive and valuable, it is unclear how generalizable the results are to other karst systems. The authors suggest that their analysis may be useful for future modeling efforts, and while they will certainly be invaluable in efforts to model this specific system, it is not clear that they can support modeling in other locations, particularly since the methods needed to reproduce this type of study elsewhere are quite costly and time-intensive. If the three source region types (D1, D2, D3) can be more clearly defined and if they can be generalized to exist in other karst systems, it would be highly beneficial if the authors provided a set of metrics that could be used to identify these types of source regions in other karst systems in the absence of high-resolution ERT surveys.

Specific comments

Abstract lines 26-30: Consistent abbreviations should be used for the three types of hydrologic source region dynamics - the main text uses D1, D2, D3, but the abstract uses (i), (ii), (iii).

Abstract line 31: The connection between the drip discharge spots and the source regions imaged using ERT methods should be made clear. Are the source region being imaged connected to specific drip discharge monitoring points, or to general types of observed drip discharge patterns? Specific examples of how this study could be used to support modeling should be provided in the main text to support this claim. It may also be worthwhile to mention the possible implications for improved understanding of speleothem formation (and therefore for paleoclimate studies) in the abstract.

Section 1 lines 37-54: The section describing the upper layers of karst systems and the associated infiltration and recharge dynamics needs to be clearer. As a reader already familiar with karst systems, I found it difficult to understand what the authors were trying to describe - I suspect it would therefore be almost incomprehensible to readers not already well-versed in the subject. This is particularly important given the unfortunately non-standard terminology used by different authors in describing karst systems above the water table. What is referred to in this paper as the "infiltration zone" (based on Mangin 1974?) is in other cases referred to as the "transmission zone" (Williams 2008) or the "unsaturated zone" (Goldscheider & Drew 2007). Additionally, because the terms "vadose zone" and "unsaturated zone" are also used in porous aquifers, it may not be clear to readers what exactly the authors mean by these terms. For example, in some texts, the terms "unsaturated zone" or "vadose zone" in karst do not include the epikarst and the soil, but in other texts, including this paper, the terms "unsaturated zone" and "vadose zone" are used to refer to everything above the water table. It would be helpful as well to choose either "unsaturated zone" or "vadose zone" and use a single term continuously throughout the paper. A simple figure could easily clarify this section - something like Fig. 2 from: Doerfliger et al. Water vulnerability assessment in karst environments: a new method of defining protection areas using a multi-attribute approach and GIS tools (EPIK method). Environmental Geology 39, 165–176 (1999) or Fig. 3 from: Bakalowicz, M. The epikarst, the skin of karst. Karst Waters Institute Special Publication 9, 16–22 (2004).

Section 1 lines 58 &74: These are important points and should be emphasized.

Section 1 section starting line 79 & section starting line 87: The order of these sections should be reversed - first ERT should be introduced, then examples of ERT studies in karst should be given.

Section 1 line 99: This is exciting and should be emphasized. It could potentially be moved to the beginning of the introduction? Also, "permanently installed" suggests that the ERT installation will be left in place and that data will continue to be collected in the

future. Is this in fact the case? If yes, it should be explicitly stated, since this will be an exciting ongoing data source.

Section 2: The description of the system is clear and helpful, and the accompanying figure is clear and provides relevant context.

Section 2 line 122: The term "decametric" is uncommon in English (it is primarily used to describe radio wavelengths). In this case, is it intended to mean that each layer is ~10 m thick? If so, specify.

Section 2 line 128: Sinkhole and doline do not always mean the same thing. In this case the formation in question appears to be more of a typical sinkhole.

Section 2 line 137: The phrase "tiny underground river" is subjective - it would be helpful to specify an estimated discharge range.

Section 2 line 144: "Detrimentally affects" is subjective. Does detrimental imply increased erosion? Damage to man-made access structures? Specify.

Figure 1a: The Lomme River is difficult to see because it is so faint and small. The blue line and text indicating the river should be thicker.

Section 3: This section is excellent - clear descriptions and extremely helpful accompanying figures.

Figure 2: There is a great deal of valuable information that is overall very well presented. The stereograms and rose diagrams especially are helpful. However, the figure would benefit from some clarification since it is so information-dense: - Include S and N indicators on the cave cross-section. - PWD3 is not shown - there should be some indication, even if only in the caption, of where it is. - Line 175 seems to discuss layers on the southern side of the cave, but the diagram only shows these layers (50-53) visible on the northern roof. - Part C is difficult to read - does the meter-ruler indicate the location of the borehole? Is there a difference between the left and right sides? What are the overlaid numbered beds meant to indicate? And how exactly are

the layers in Part C connected to the layers in the cave shown in Part A? See comments on Fig. 15 as well. - In the caption, spell out that S0 refers to bedding planes.

Section 4: Fig. 1 seems to show steps descending into the sinkhole. This should be mentioned in the site description. Were the steps constructed for this study? Does the general public have access to the interior of the sinkhole?

Section 4.1 lines 220-221: Clarify how humidity data will contribute to understanding infiltration dynamics, or move this sentence to right before line 269.

Section 4.2 line 226: Some indication of what the normal climate and precipitation patterns for this region are should be made earlier, in the site description.

Section 4.2 line 266: At some point before discussion of ERT results and/or expected patterns, there should be a brief description of what factors generally increase resistivity, and what factors generally decrease resistivity, so that readers not familiar with ERT methods have a frame of reference for this type of statement (see comments on Section 7.2 as well).

Section 4.2 line 283: Define "shaft flows".

Section 4.2 line 290: Clarify that PWD1 and PWD2 and spatially located close together, not close in terms of similar flow patterns.

Section 4.2 lines 307-314: Emphasize this section.

Section 4.2 line 326: Make a brief summary of the primary findings of the study (one sentence).

Section 5: I am not very familiar with ERT methods, and so did not give detailed comments for this section.

Figure 6: The caption for this figure should include what the letters (A-D) indicate, and what the primary differences between DD and GD datasets are. It should also spell out DD and GD rather than using abbreviations.

Figure 7: The red lines and numbers indicating where in time each resistivity image belongs are extremely helpful. These data could be powerfully visualized using a simple time-lapse animation (a GIF would work well) of the resistivities. This can be done very easily with several freely available software options (see one example of the step-by-step process here: http://www.spacelapse.net/en/Astrophotography_Tutorials/Convert_Single_Photos_to_a_Timelapse_Movie.html). Such an animation could be posted as supplementary material when the article is published online, and would give a compelling image of the subsurface dynamics over time.

Section 7.1 lines 575: Include possible options for distinguishing between the two types.

Figure 8: Very nice. Include a brief description of the characteristics of each region in the caption.

Figure 10: This is an excellent figure - very clear and detailed. Include descriptions of each numbered region in the caption.

Section 7.2 line 640-641: Something to this effect should be explained much much earlier in the paper, in the introduction.

Section 7.2.1 line 691: This sentence needs to be reversed - the increased pore water conductivity is a result of the drought, not the opposite.

Section 7.2.1 lines 699-707: Is it possible to disentangle the effects of fresh rainwater mixing from the effects of increased saturation? If not, how significant is the uncertainty introduced by the different conductivities of rain & pore waters?

Section 7.2.2 line 734: Is the fracture drip discharge at PWD1 from a different fracture than the one identified az zone #8 on Fig. 10? It is important to spell this out. Is it assumed that if drip discharge could be monitored from the #8 fracture zone, it would follow a similar pattern to PWD1? The descriptions make it unclear how D1 type

regions differ from D3 type regions - they are both fractured regions? Is D3 intermediate between D1 (quick response to precip) and D2 (damped response to precip)? Why does PWD1 correlate with D1 type if it is measuring fracture drip discharge? One would think it should correlated with D3 type? In the text, PWD1 is listed as correlated with D1 regions, but in Fig. 15, PWD1 is shown as correlated with the D3 fractured region. Which is correct? Also, it is possible to estimate rates of recharge for each different type of region? Is it thought that these three types exist in other karst systems as well?

Section 7.2.2 line 739: Clarify the use of the term "subsurface." Is this intended to indicate near-surface regions? It is more commonly used to indicate everything below the surface, so the meaning should be made explicit.

Section 7.2.2 line 751: This type of general principle for interpreting ERT data should be listed out in one section nearer the beginning of the paper, or at the very start of the analysis section.

Section 7.2.2 line 778: Again, is the drip discharge at PWD2 & PWD3 coming from the same layers as those identified by ERT analysis as type D2? Or are they following similar behavior patterns but not coming from the same units? This should be explicitly stated.

Section 7.2.2 line 791-797: In this paragraph, does "the fractured region" correspond to D3 type? And does "surface layers" correspond to D1 type? Is the primary difference between D1 and D3 that the response to rainfall is more short-lived in D3?

Section 7.2.3 line 875: The preceding section seems to reject both possible causes of the increased resistivity after rainfall. What might then be responsible?

Section 7.3 line 886: The previous sentences indicate that the epikarst does not act as a buffer, but this sentence states that the epikarst does act as a buffer. Clarify.

Figure 15: This is an excellent overview/summary figure. The seasonal variation indicators are particularly helpful. However, in combination with Fig. 2 it is confusing. Fig. 2 seems to indicate that the ERT transect is next to the cave, but Fig. 15 seems to indicate that the cave is directly below the ERT transect. Which is correct? Both figures should be revised to be consistent with each other. The caption for Fig. 15 should also be much more detailed, with brief summaries of the D1, D2, and D3 dynamics, and the PWD1, 2, and 3 dynamics.

Section 8: This section is overall clear and concise. The phrasing in this section could be adapted to clarify some of the previous sections.

Section 8 lines 939-943: It would be good to think about how the interpretations drawn from ERT data could be further tested hydrologically. Are there other possible interpretations? Could different interpretations be tested against each other? What additional data could support or counter these interpretations?

Technical corrections

There are many small typos, spelling errors, and grammar issues. This paper would benefit from a purely copy-editing oriented revision. A few of the most obvious ones are listed below:

l. 17 carbonate l. 39 soil joined with the epikarst l. 54 agree on the dichotomy of matrix and conduit l. 114 the Charlesmont Limestone, which includes l. 115 member acts as l. 117 system ends when l. 189 three groups of joint orientations l. 251 Penman-Monteith l. 256 homogeneously l. 534 inflection l. 740 Klimchouk (2004), defining l. 787 This indicates delayed

Please also note the supplement to this comment:
https://www.hydrol-earth-syst-sci-discuss.net/hess-2017-477/hess-2017-477-RC1-supplement.pdf
* * *
477, 2017.

**Supplement:**

**HESS Review Criteria**

| Principal criteria | Excellent (1) | Good (2) | Fair (3) | Poor (3) |
|---|---|---|---|---|
| **Scientific significance:** Does the manuscript represent a substantial contribution to scientific progress within the scope of Hydrology and Earth System Sciences (substantial new concepts, ideas, methods, or data)? | | Substantial and impressive new datasets, and broadens applications of ERT methods. | | |
| **Scientific quality:** Are the scientific approach and applied methods valid? Are the results discussed in an appropriate and balanced way (consideration of related work, including appropriate references)? | | I am not familiar with ERT methods, but the presentation and discussion of methods, data, and results felt appropriate, though at times difficult to understand without detailed prior knowledge of the method. | | |
| **Presentation quality:** Are the scientific results and conclusions presented in a clear, concise, and well-structured way (number and quality of figures/tables, appropriate use of English language)? | | | Writing has many small spelling/grammar/typo issues, and structure of sentences and paragraphs is often not clear. This paper could probably be much shorter. | |

Does the paper address relevant scientific questions within the scope of HESS?
*Yes. Seasonal variation in subsurface water flow processes in complex karst systems.*

Does the paper present novel concepts, ideas, tools, or data?
*Yes. Significant, comprehensive, multi-year, multi-instrument dataset, and expanded applications of ERT methods. The data will be made freely available.*

Are substantial conclusions reached?
*Somewhat. The ERT data supported a commonly-accepted model of karst aquifer functioning, and provided detailed information describing the flow patterns sustaining cave drip discharge. The authors hope that this data will support modeling, but it is not clear whether enough generalizable information is provided to support modeling in other karst systems or only in this specific location. If a semi-permanent ERT installation is needed to detect preferential flow paths in every aquifer we wish to model, the costs are likely too high to make it feasible.*

Are the scientific methods and assumptions valid and clearly outlined?
*Based on my limited familiarity with ERT methods, yes.*

Are the results sufficient to support the interpretations and conclusions?
*Based on my limited familiarity with ERT methods, yes.*

Is the description of experiments and calculations sufficiently complete and precise to allow their reproduction by fellow scientists (traceability of results)?

*Based on my limited familiarity with ERT methods, yes.*

Do the authors give proper credit to related work and clearly indicate their own new/original contribution?

*Based on my knowledge of the literature, yes.*

Does the title clearly reflect the contents of the paper?

*Yes.*

Does the abstract provide a concise and complete summary?

*Yes.*

Is the overall presentation well structured and clear?

*Somewhat. The paper is quite long, and would therefore benefit from a clearer organizational structure to emphasize each section's key points. The writing is also sometimes convoluted, and had many spelling & grammar errors and typos.*

Is the language fluent and precise?

*Somewhat. Many sentences would benefit from being streamlined, and in some cases the sentence structure implies things that the authors probably did not intend.*

Are mathematical formulae, symbols, abbreviations, and units correctly defined and used?

*Yes.*

Should any parts of the paper (text, formulae, figures, tables) be clarified, reduced, combined, or eliminated?

*The text would benefit from a thorough copy-edit.*

*The introduction in particular uses needlessly complex language to describe the basic functioning of karst aquifers and the role of the epikarst. An additional figure here would probably help.*

*The figures are generally very strong.*

*A few clarifying edits are suggested, particularly for Fig. 2 and Fig. 15.*

*For readers not familiar with ERT methods, a very brief introduction to the general concept and to the factors affecting resistivity (low resistivity = higher conductivity = higher permeability, higher water content, more conductive material...etc.) would be helpful.*

Are the number and quality of references appropriate?

*Yes.*

Is the amount and quality of supplementary material appropriate?

*Yes.*

**Comment:**

**General comments**

This review will focus on the structure, context, and implications of the paper with respect to karst hydrogeology, rather than on specific comments on ERT methods and analysis, as I am not well-acquainted with ERT techniques.

This paper presents and makes freely available valuable data from long-term, high-resolution geophysical monitoring of groundwater flow patterns in a karst system. These data support well-accepted conceptual models for infiltration and recharge processes in karst systems, particularly with respect to the role of the epikarst. The authors do not put forward new conceptual models, but present the data and their analyses and conclusions in high-quality, detailed, well-developed figures, which are a major highlight of the paper. However, the overall quality of the writing and the organization of the paper lack clarity and concision. The paper would benefit from thorough, streamlining-focused editing, and from clearer explanations of the underlying concepts and assumptions the study is

based on. Additionally, the authors propose one specific interpretation of the ERT data, but their analysis might benefit from considering what other interpretations would be possible, and what hydrogeologic data would be needed to test competing interpretations against each other and against additional data. Finally, although the dataset being presented is extremely impressive and valuable, it is unclear how generalizable the results are to other karst systems. The authors suggest that their analysis may be useful for future modeling efforts, and while they will certainly be invaluable in efforts to model this specific system, it is not clear that they can support modeling in other locations, particularly since the methods needed to reproduce this type of study elsewhere are quite costly and time-intensive. If the three source region types (D1, D2, D3) can be more clearly defined and if they can be generalized to exist in other karst systems, it would be highly beneficial if the authors provided a set of metrics that could be used to identify these types of source regions in other karst systems in the absence of high-resolution ERT surveys.

**Specific comments**

| Location | Comment |
|---|---|
| Abstract
lines 26-30 | Consistent abbreviations should be used for the three types of hydrologic source region dynamics - the main text uses D1, D2, D3, but the abstract uses (i), (ii), (iii). |
| Abstract
line 31 | The connection between the drip discharge spots and the source regions imaged using ERT methods should be made clear. Are the source region being imaged connected to specific drip discharge monitoring points, or to general types of observed drip discharge patterns?

Specific examples of how this study could be used to support modeling should be provided in the main text to support this claim.

It may also be worthwhile to mention the possible implications for improved understanding of speleothem formation (and therefore for paleoclimate studies) in the abstract. |
| Section 1
lines 37-54 | The section describing the upper layers of karst systems and the associated infiltration and recharge dynamics needs to be clearer. As a reader already familiar with karst systems, I found it difficult to understand what the authors were trying to describe - I suspect it would therefore be almost incomprehensible to readers not already well-versed in the subject. This is particularly important given the unfortunately non-standard terminology used by different authors in describing karst systems above the water table. What is referred to in this paper as the "infiltration zone" (based on Mangin 1974?) is in other cases referred to as the "transmission zone" (Williams 2008) or the "unsaturated zone" (Goldscheider & Drew 2007). Additionally, because the terms "vadose zone" and "unsaturated zone" are also used in porous aquifers, it may not be clear to readers what exactly the authors mean by these terms. For example, in some texts, the terms "unsaturated zone" or "vadose zone" in karst *do not include the epikarst and the soil*, but in other texts, including this paper, the terms "unsaturated zone" and "vadose zone" are used to refer to everything above the water table. It would be helpful as well to choose either "unsaturated zone" or "vadose zone" and use a single term continuously throughout the paper.
A simple figure could easily clarify this section - something like Fig. 2 from: Doerfliger et al. Water vulnerability assessment in karst environments: a new method of defining protection areas using a multi-attribute approach and GIS tools (EPIK method). *Environmental Geology* **39,** 165–176 (1999)
or Fig. 3 from:
Bakalowicz, M. The epikarst, the skin of karst. *Karst Waters Institute Special* |

| | |
|---|---|
| | *Publication* **9,** 16–22 (2004). |
| Section 1
lines 58 &74 | These are important points and should be emphasized. |
| Section 1
section starting line 79
& section starting line 87 | The order of these sections should be reversed - first ERT should be introduced, then examples of ERT studies in karst should be given. |
| Section 1
line 99 | This is exciting and should be emphasized. It could potentially be moved to the beginning of the introduction?
Also, "permanently installed" suggests that the ERT installation will be left in place and that data will continue to be collected in the future. Is this in fact the case? If yes, it should be explicitly stated, since this will be an exciting ongoing data source. |
| Section 2 | The description of the system is clear and helpful, and the accompanying figure is clear and provides relevant context. |
| Section 2
line 122 | The term "decametric" is uncommon in English (it is primarily used to describe radio wavelengths). In this case, is it intended to mean that each layer is ~10 m thick? If so, specify. |
| Section 2
line 128 | Sinkhole and doline do not always mean the same thing. In this case the formation in question appears to be more of a typical sinkhole. |
| Section 2
line 137 | The phrase "tiny underground river" is subjective - it would be helpful to specify an estimated discharge range. |
| Section 2
line 144 | "Detrimentally affects" is subjective. Does detrimental imply increased erosion? Damage to man-made access structures? Specify. |
| Figure 1a | The Lomme River is difficult to see because it is so faint and small. The blue line and text indicating the river should be thicker. |
| Section 3 | This section is excellent - clear descriptions and extremely helpful accompanying figures. |
| Figure 2 | There is a great deal of valuable information that is overall very well presented. The stereograms and rose diagrams especially are helpful. However, the figure would benefit from some clarification since it is so information-dense:
- Include S and N indicators on the cave cross-section.
- PWD3 is not shown - there should be some indication, even if only in the caption, of where it is.
- Line 175 seems to discuss layers on the southern side of the cave, but the diagram only shows these layers (50-53) visible on the northern roof.
- Part C is difficult to read - does the meter-ruler indicate the location of the borehole? Is there a difference between the left and right sides? What are the overlaid numbered beds meant to indicate? And how exactly are the layers in Part C connected to the layers in the cave shown in Part A? See comments on Fig. 15 as well.
- In the caption, spell out that S0 refers to bedding planes. |
| Section 4 | Fig. 1 seems to show steps descending into the sinkhole. This should be mentioned |

| | |
|---|---|
| | in the site description. Were the steps constructed for this study? Does the general public have access to the interior of the sinkhole? |
| Section 4.1
lines 220-221 | Clarify how humidity data will contribute to understanding infiltration dynamics, or move this sentence to right before line 269. |
| Section 4.2
line 226 | Some indication of what the normal climate and precipitation patterns for this region are should be made earlier, in the site description. |
| Section 4.2
line 266 | At some point *before* discussion of ERT results and/or expected patterns, there should be a brief description of what factors generally increase resistivity, and what factors generally decrease resistivity, so that readers not familiar with ERT methods have a frame of reference for this type of statement (see comments on Section 7.2 as well). |
| Section 4.2
line 283 | Define "shaft flows". |
| Section 4.2
line 290 | Clarify that PWD1 and PWD2 and spatially located close together, not close in terms of similar flow patterns. |
| Section 4.2
lines 307-314 | Emphasize this section. |
| Section 4.2
line 326 | Make a *brief* summary of the primary findings of the study (one sentence). |
| Section 5 | I am not very familiar with ERT methods, and so did not give detailed comments for this section. |
| Figure 6 | The caption for this figure should include what the letters (A-D) indicate, and what the primary differences between DD and GD datasets are. It should also spell out DD and GD rather than using abbreviations. |
| Figure 7 | The red lines and numbers indicating where in time each resistivity image belongs are extremely helpful.
These data could be powerfully visualized using a simple time-lapse animation (a GIF would work well) of the resistivities. This can be done very easily with several freely available software options (see one example of the step-by-step process here: http://www.spacelapse.net/en/Astrophotography_Tutorials/Convert_Single_Photos_to_a_Timelapse_Movie.html). Such an animation could be posted as supplementary material when the article is published online, and would give a compelling image of the subsurface dynamics over time. |
| Section 7.1
lines 575 | Include possible options for distinguishing between the two types. |
| Figure 8 | Very nice. Include a brief description of the characteristics of each region in the caption. |
| Figure 10 | This is an excellent figure - very clear and detailed. Include descriptions of each numbered region in the caption. |

| Section 7.2
line 640-641 | Something to this effect should be explained much much earlier in the paper, in the introduction. |
|---|---|
| Section 7.2.1
line 691 | This sentence needs to be reversed - the increased pore water conductivity is a result of the drought, not the opposite. |
| Section 7.2.1
lines 699-707 | Is it possible to disentangle the effects of fresh rainwater mixing from the effects of increased saturation? If not, how significant is the uncertainty introduced by the different conductivities of rain & pore waters? |
| Section 7.2.2
line 734 | Is the fracture drip discharge at PWD1 from a different fracture than the one identified az zone #8 on Fig. 10? It is important to spell this out. Is it assumed that if drip discharge could be monitored from the #8 fracture zone, it would follow a similar pattern to PWD1? The descriptions make it unclear how D1 type regions differ from D3 type regions - they are both fractured regions? Is D3 intermediate between D1 (quick response to precip) and D2 (damped response to precip)? Why does PWD1 correlate with D1 type if it is measuring fracture drip discharge? One would think it should correlated with D3 type? In the text, PWD1 is listed as correlated with D1 regions, but in Fig. 15, PWD1 is shown as correlated with the D3 fractured region. Which is correct?

Also, it is possible to estimate *rates* of recharge for each different type of region? Is it thought that these three types exist in other karst systems as well? |
| Section 7.2.2
line 739 | Clarify the use of the term "subsurface." Is this intended to indicate near-surface regions? It is more commonly used to indicate everything below the surface, so the meaning should be made explicit. |
| Section 7.2.2
line 751 | This type of general principle for interpreting ERT data should be listed out in one section nearer the beginning of the paper, or at the very start of the analysis section. |
| Section 7.2.2
line 778 | Again, is the drip discharge at PWD2 & PWD3 coming from the same layers as those identified by ERT analysis as type D2? Or are they following similar behavior patterns but not coming from the same units? This should be explicitly stated. |
| Section 7.2.2
line 791-797 | In this paragraph, does "the fractured region" correspond to D3 type? And does "surface layers" correspond to D1 type? Is the primary difference between D1 and D3 that the response to rainfall is more short-lived in D3? |
| Section 7.2.3
line 875 | The preceding section seems to reject both possible causes of the increased resistivity after rainfall. What might then be responsible? |
| Section 7.3
line 886 | The previous sentences indicate that the epikarst does *not* act as a buffer, but this sentence states that the epikarst *does* act as a buffer. Clarify. |
| Figure 15 | This is an excellent overview/summary figure. The seasonal variation indicators are particularly helpful.
However, in combination with Fig. 2 it is confusing. Fig. 2 seems to indicate that the ERT transect is next to the cave, but Fig. 15 seems to indicate that the cave is directly below the ERT transect. Which is correct? Both figures should be revised to be consistent with each other.
The caption for Fig. 15 should also be much more detailed, with brief summaries of the D1, D2, and D3 dynamics, and the PWD1, 2, and 3 dynamics. |

| Section 8 | This section is overall clear and concise. The phrasing in this section could be adapted to clarify some of the previous sections. |
|---|---|
| Section 8 lines 939-943 | It would be good to think about how the interpretations drawn from ERT data could be further tested hydrologically. Are there other possible interpretations? Could different interpretations be tested against each other? What additional data could support or counter these interpretations? |

**Technical corrections**

There are many small typos, spelling errors, and grammar issues. This paper would benefit from a purely copy-editing oriented revision. A few of the most obvious ones are listed below:

l. 17  carbonate
l. 39  soil joined with the epikarst
l. 54  agree on the dichotomy of matrix and conduit
l. 114 the Charlesmont Limestone, which includes
l. 115 member acts as
l. 117 system ends when
l. 189 three groups of joint orientations
l. 251 Penman-Monteith
l. 256 homogeneously
l. 534 inflection
l. 740 Klimchouk (2004), defining
l. 787 This indicates delayed

---

## Editor Comment (EC1) · T. P. A. Ferre (Editor) · 6 Dec 2017

The authors have addressed a particularly difficult topic within hydrogeology: the dynamics of water flow in complex, layered, heterogeneous regions including relatively large preferential (karstic) pathways. They have applied an accepted and widely used hydrogeophysical method (electrical resistivity tomography) to this task. The strength of the study lies in the unique long-term (three year) data set in a karstic environment. The challenge, not surprisingly given the complexity of the system, lies in the interpretation of hydrogeophysical data and the transferrability of the method and results.

As is fitting for an exploratory application of a new method, the study was conducted in a very well characterized area. It would be worthwhile commenting on how well

the ERT data could have been interpreted in the absence of this additional data. This has direct relevance to the use of the ERT method for other, less well-characterized sites. Additionally, the system under study is particularly amenable to study because the water table traverses the known conduits in a typical year, flooding them during winter and running dry in summer. This, again, indicates a wise choice of method for a specific study area. But, it would be worthwhile to comment on this specifically when suggesting that the ERT method could be useful at other karst sites. In other words, it is well known that ERT can only monitor dynamics in as much as it identifies temporal changes in water saturation. How can a reader decide if those conditions are likely to exist at measurable levels at a site before deciding to conduct an ERT survey?

The ERT surveys appear to have been very well designed. The combined use of DD and GD surveys is thoughtful and the description of their differences in terms of spatial information and practical limitations is succinctly stated. The use of automated data collection and analysis, incorporating reciprocal measurements where available, gives confidence in the data quality. Similarly, the authors' recognition of temporal variations in contact resistance is noted as this is often overlooked in long term studies.

I was impressed by the approach used to correct for temperature effects. However, it isn't clear to me that the heat transport was coupled with water flow. Given the complexity of the hydrologic system, this may not have been possible. But, it would be good to add more detail regarding how the temperature distribution was determined to allow for temperature correction. (This may be suited to an appendix.)

It is not clear to me if the resistivity inversion is spatio-temporal or if each time slice was inverted separately. Given that you are looking for changes in time, it seems that spatio-temporal inversion may be more appropriate. But, I would like to have seen some discussion of this choice. It also strikes me that interpreting each time slice independently may be more subject to imposing small scale seasonal variations in areas that are actually not seeing any real variation. It would be very interesting to see if a time lapse inversion results in as good a fit with some areas showing no seasonal

EC changes.

In the end, I felt that the strongest element of this paper was the structural interpretation. This would be strengthened by more discussion of the process by which the arbitrary conductivity-bounds between regions were determined. It would be far more useful if this were explored automatically, perhaps using clustering techniques to propose alternative structural maps. For me, I think that some discussion of the EC limits is needed for publication. But, the paper would have more impact if this analysis were expanded and potentially seen as the basis for forming competing structural hypotheses. The danger as presented is that the authors may have unconsciously chosen EC limits to confirm their pre-existing structural interpretation. This would, of course, limit the value of all of the work that went into collecting the data.

Unfortunately, and not unexpectedly given the complexity of the system under study, I found that the hydrologic interpretations were somewhat qualitative. It is interesting to see that there are correlations and delays between responses. But, it doesn't seem to rise to the level of increased understanding of flow dynamics. This may simply be a matter of emphasis - you could highlight what was learned from the ERT that would not have been possible without it. But, it reads more like using your hydrologic insight to give plausible explanations of what you see in the ERT results. Understand, this isn't a strong criticism. I think that this is an advance and shows potential future use of ERT for monitoring dynamics in some karstic systems. But, I think that it is a mistake to make this the emphasis of the paper - starting with the title. Rather, I would focus the paper on the 'hydrostratigraphic' results - showing how you could use the time lapse ERT to identify structure in the subsurface. That could be expanded and extended and then it would be appropriate to say that this interpretation is consistent with what was seen in other hydrologic measurements. As an added benefit, this would allow you to shrink the hydrologic section, which is not as tightly written as the previous sections.

In summary, I think that this is a very strong paper and that it should be published in HESS. But, I think that the current emphasis on flow dynamics is not ideal. Rather,

it could be a very novel and interesting example of using dynamics to better define structure. This would be most interesting if it could be done automatically, e.g. using clustering tools, and if it led to multiple competing hypotheses that could be further tested in the field.

---

## Author Comment (AC1) · 20 Dec 2017

We thank Referee #1 for his detailed and relevant comments that will improve the overall quality of the manuscript. Our answers (A) to the comments (R) can be find below.

R: This review will focus on the structure, context, and implications of the paper with respect to karst hydrogeology, rather than on specific comments on ERT methods and analysis, as I am not well-acquainted with ERT techniques. This paper presents and makes freely available valuable data from long-term, high-resolution geophysical monitoring of groundwater flow patterns in a karst system. These data support well-accepted conceptual models for infiltration and recharge processes in karst systems,

[Figure]

particularly with respect to the role of the epikarst. The authors do not put forward new conceptual models, but present the data and their analyses and conclusions in high-quality, detailed, well-developed figures, which are a major highlight of the paper. However, the overall quality of the writing and the organization of the paper lack clarity and concision. The paper would benefit from thorough, streamlining-focused editing, and from clearer explanations of the underlying concepts and assumptions the study is based on. Additionally, the authors propose one specific interpretation of the ERT data, but their analysis might benefit from considering what other interpretations would be possible, and what hydrogeologic data would be needed to test competing interpretations against each other and against additional data. Finally, although the dataset being presented is extremely impressive and valuable, it is unclear how generalizable the results are to other karst systems. The authors suggest that their analysis may be useful for future modeling efforts, and while they will certainly be invaluable in efforts to model this specific system, it is not clear that they can support modeling in other locations, particularly since the methods needed to reproduce this type of study elsewhere are quite costly and time-intensive. If the three source region types (D1, D2, D3) can be more clearly defined and if they can be generalized to exist in other karst systems, it would be highly beneficial if the authors provided a set of metrics that could be used to identify these types of source regions in other karst systems in the absence of high-resolution ERT surveys.

A: Referee #1 has built his comments with respect to karst hydrogeology, which is much appreciated by the authors. It raises two main criticisms on the paper: 1) the lack of proposed alternative hydrogeological interpretations, and 2) a general concern about how our results could be of benefit to studies of other karst systems. He also suggests rearranging the organization and writing of the paper. The comments point some fragilities of the current version of the manuscript. We will therefore rearrange the manuscript to clarify these points.

As for the main concerns of Referee #1, we propose one main hydrogeological interpretation, because it is supported by different independent sources of information, i.e. drip discharge data, ERT results, soil moisture data and meteorological data. We have built this experiment as a multi-method approach because this karst system, as many others, is complex and could not be studied in details with one single method only. This paper demonstrates that when studying such a complex system, using multi-method monitoring is required to investigate the groundwater recharge. Hopefully, we believe that with such an amount of data, collected within 3 hydrological cycles, we can propose one hydrological interpretation, strong enough that there is no need to propose alternative interpretations which would require further measurements to be validated. However the results leading to this interpretation are discussed in details within several sections of the manuscript. Especially, we demonstrate the applicability of combining time-lapse ERT and drip discharge monitoring, which is new and can be applied in other karst systems. To enhance this aspect, we are working on a future paper in which we will focus on a lumped karst modeling of the vadose zone infiltration processes based on the drip discharge data (using the KarstMod modeling platform, from Mazzilli et al., 2017, DOI: 10.1016/j.envsoft.2017.03.015). The preliminary results support the interpretations drawn with the ERT monitoring, such as PWD1 sampling a small soil/epikarst reservoir. Simulations on PWD2 data show that a larger epikarst reservoir is responsible for the seasonal variations. A hysteretic function explains the deactivation of the drip discharge in dry periods. We could relate this to spatial disconnection in the ERT images between the superficial layers and deeper regions such as the porous limestone region imaged in the ERT profile. The strength of the ERT monitoring therefore lies in its applicability to image and spatialize conceptual objects defined within karst modeling. In any cases, such types of study highlight the strong link between ERT monitoring and karst modeling aspects as we claim in the abstract, while it strengthen a possible approach to be tested in other karst systems. We will definitely add some details on these future research opportunities in the concluding remarks section. Moreover, to account for Referee #1's concerns, we will add some words on what methods could be alternatively tested in the future to challenge and/or confirm our

hydrological interpretations. For example, passive seismic noise monitoring networks have recently proved their applicability to track groundwater content variations at several depths (e.g. Voisin et al. 2016, DOI: 10.1190/INT-2016-0010.1). Such geophysical techniques could bring additional and independent sources of information to compare with the ERT monitoring results.

Referee #1 also suggests providing metrics to detect in other karst systems the types of source regions (D1, D2, D3) defined within this study. One important message of our study concerns the fact that we strongly suggest complementing any ERT experiment conducted in a karst area with geological and structural investigations of the studied site. This is especially true because ERT monitoring is meant to identify infiltration dynamics to be correlated with lithological and structural information, in order to build models at larger scale than that of the ERT monitoring site itself. The relationships between geological settings and infiltration dynamics are subject to vary from one karst to another. Nonetheless, some of the interpretations drawn in our study are transferable to other karst systems. For example, superficial areas are likely to exhibit resistivity dynamics close to the D1 type defined in our study, with rapid changes in response to rainfall events. This is likely to be one signature of the epikarst layer. At greater depths, strong response to rainfall would rather suggest fractured regions as well. We could therefore highlight some clues in the concluding remarks that could be used in other karst systems to quickly identify those types of dynamics, to be validated via geological investigations.

Below, we respond to all the specific comments of Referee #1.

Specific comments

R: Abstract lines 26-30: Consistent abbreviations should be used for the three types of hydrologic source region dynamics - the main text uses D1, D2, D3, but the abstract uses (i), (ii), (iii).

A: This will be updated in the abstract.

R: Abstract line 31: The connection between the drip discharge spots and the source regions imaged using ERT methods should be made clear. Are the source region being imaged connected to specific drip discharge monitoring points, or to general types of observed drip discharge patterns? Specific examples of how this study could be used to support modeling should be provided in the main text to support this claim. It may also be worthwhile to mention the possible implications for improved understanding of speleothem formation (and therefore for paleoclimate studies) in the abstract.

A: As shown in Figure 2 and summarized in Figure 15, the three drip discharge spots do not sample the same strata/fractures as those being sampled by the ERT surveys. We are therefore pointing to general types of observed drip discharge patterns, as explained in the discussion section. It should however be made clearer in the main text from the beginning. As for the support to modeling approaches, a paragraph will be added in the discussion to strengthen this aspect. It will mainly focus on the way ERT measurements can help the modeling of reservoir discharges, as said in response to the general comment.

R: Section 1 lines 37-54: The section describing the upper layers of karst systems and the associated infiltration and recharge dynamics needs to be clearer. As a reader already familiar with karst systems, I found it difficult to understand what the authors were trying to describe - I suspect it would therefore be almost incomprehensible to readers not already well-versed in the subject. This is particularly important given the unfortunately non-standard terminology used by different authors in describing karst systems above the water table. What is referred to in this paper as the "infiltration zone" (based on Mangin 1974?) is in other cases referred to as the "transmission zone" (Williams 2008) or the "unsaturated zone" (Goldscheider & Drew 2007). Additionally, because the terms "vadose zone" and "unsaturated zone" are also used in porous aquifers, it may not be clear to readers what exactly the authors mean by these terms. For example, in some texts, the terms "unsaturated zone" or "vadose zone" in karst do not include the epikarst and the soil, but in other texts, including this paper, the terms

"unsaturated zone" and "vadose zone" are used to refer to everything above the water table. It would be helpful as well to choose either "unsaturated zone" or "vadose zone" and use a single term continuously throughout the paper. A simple figure could easily clarify this section - something like Fig. 2 from: Doerfliger et al. Water vulnerability assessment in karst environments: a new method of defining protection areas using a multi-attribute approach and GIS tools (EPIK method). Environmental Geology 39, 165–176 (1999) or Fig. 3 from: Bakalowicz, M. The epikarst, the skin of karst. Karst Waters Institute Special Publication 9, 16–22 (2004).

A: We will rearrange the introduction, so that a paragraph will clearly describe what terminology we are using in this paper. As for the choice between unsaturated and vadose zone, we will change the only occurrence of unsaturated zone in the manuscript to vadose zone. As suggested by Referee #1, we will include a figure to clarify this section.

R: Section 1 lines 58 &74: These are important points and should be emphasized. A: This will be emphasized.

R: Section 1 section starting line 79 & section starting line 87: The order of these sections should be reversed - first ERT should be introduced, then examples of ERT studies in karst should be given.

A: Paragraph will be inversed.

R: Section 1 line 99: This is exciting and should be emphasized. It could potentially be moved to the beginning of the introduction? Also, "permanently installed" suggests that the ERT installation will be left in place and that data will continue to be collected in the future. Is this in fact the case? If yes, it should be explicitly stated, since this will be an exciting ongoing data source.

A: This will be emphasized in the introduction. And yes, the data acquisition is planned to continue in the future.

R: Section 2: The description of the system is clear and helpful, and the accompanying figure is clear and provides relevant context.

A: Thanks.

R: Section 2 line 122: The term "decametric" is uncommon in English (it is primarily used to describe radio wavelengths). In this case, is it intended to mean that each layer is âĹij 10 m thick? If so, specify.

A: This term is indeed misused. We rather point to "series of decimetric layers".

R: Section 2 line 128: Sinkhole and doline do not always mean the same thing. In this case the formation in question appears to be more of a typical sinkhole.

A: We will only use sinkhole throughout the paper.

R: Section 2 line 137: The phrase "tiny underground river" is subjective - it would be helpful to specify an estimated discharge range.

A: The discharge range of the underground river will be specified.

R: Section 2 line 144: "Detrimentally affects" is subjective. Does detrimental imply increased erosion? Damage to man-made access structures? Specify.

A: We intend to mean that a significant portion of the caves is not affected by the flash flood in the RCL area, which is at the center of the Lorette Cave. We will rephrase this in the text.

R: Figure 1a: The Lomme River is difficult to see because it is so faint and small. The blue line and text indicating the river should be thicker.

A: We will modify the figure.

R: Section 3: This section is excellent - clear descriptions and extremely helpful accompanying figures.

A: Thanks!

R: Figure 2: There is a great deal of valuable information that is overall very well presented. The stereograms and rose diagrams especially are helpful. However, the figure would benefit from some clarification since it is so information-dense: - Include S and N indicators on the cave cross-section. - PWD3 is not shown - there should be some indication, even if only in the caption, of where it is. - Line 175 seems to discuss layers on the southern side of the cave, but the diagram only shows these layers (50-53) visible on the northern roof. - Part C is difficult to read - does the meter-ruler indicate the location of the borehole? Is there a difference between the left and right sides? What are the overlaid numbered beds meant to indicate? And how exactly are the layers in Part C connected to the layers in the cave shown in Part A? See comments on Fig. 15 as well. - In the caption, spell out that S0 refers to bedding planes.

A: The figure will be clarified following these suggestions, while the main text will describe in more details the figure.

R: Section 4: Fig. 1 seems to show steps descending into the sinkhole. This should be mentioned in the site description. Were the steps constructed for this study? Does the general public have access to the interior of the sinkhole?

A: We will include a short explanation about the steps and the infrastructures that were originally built for a touristic exploitation of the Cave in the beginning of the 20th century, but stopped later. The steps have been secured against collapse for our study.

R: Section 4.1 lines 220-221: Clarify how humidity data will contribute to understanding infiltration dynamics, or move this sentence to right before line 269.

A: This will be moved right before line 269, as suggested.

R: Section 4.2 line 226: Some indication of what the normal climate and precipitation patterns for this region are should be made earlier, in the site description. A: The site description will be improved in this way.

R: Section 4.2 line 266: At some point before discussion of ERT results and/or ex-

pected patterns, there should be a brief description of what factors generally increase resistivity, and what factors generally decrease resistivity, so that readers not familiar with ERT methods have a frame of reference for this type of statement (see comments on Section 7.2 as well).

A: This will be done in the introduction.

R: Section 4.2 line 283: Define "shaft flows".

A: It will be defined.

R: Section 4.2 line 290: Clarify that PWD1 and PWD2 and spatially located close together, not close in terms of similar flow patterns.

A: Indeed, this will be made clearer.

R: Section 4.2 lines 307-314: Emphasize this section.

A: It will be emphasized.

R: Section 4.2 line 326: Make a brief summary of the primary findings of the study (one sentence).

A: Good idea, it will be added.

R: Section 5: I am not very familiar with ERT methods, and so did not give detailed comments for this section.

A: OK.

R: Figure 6: The caption for this figure should include what the letters (A-D) indicate, and what the primary differences between DD and GD datasets are. It should also spell out DD and GD rather than using abbreviations.

A: OK, it will be included in the next version of the manuscript.

R: Figure 7: The red lines and numbers indicating where in time each resistivity image belongs are extremely helpful. These data could be powerfully visualized using a simple time-lapse animation (a GIF would work well) of the resistivities. This can be done very easily with several freely available software options (see one example of the step-by-step process here: http://www.spacelapse.net/en/Astrophotography_Tutorials/Convert_Single_Photos_to_a_Timelapse_Movie.html). Such an animation could be posted as supplementary material when the article is published online, and would give a compelling image of the subsurface dynamics over time. Section 7.1 lines 575: Include possible options for distinguishing between the two types.

A: Excellent idea. An animation will be added as supplementary material.

R: Figure 8: Very nice. Include a brief description of the characteristics of each region in the caption.

A: Good idea. It will be included.

R: Figure 10: This is an excellent figure - very clear and detailed. Include descriptions of each numbered region in the caption.

A: This will also be added to the figure.

R: Section 7.2 line 640-641: Something to this effect should be explained much earlier in the paper, in the introduction.

A: This is actually already explained throughout Section 5. However, we will add a clear explanation on this at the beginning of the description of the ERT results.

R: Section 7.2.1 line 691: This sentence needs to be reversed - the increased pore water conductivity is a result of the drought, not the opposite.

A: The sentence will be reversed.

R: Section 7.2.1 lines 699-707: Is it possible to disentangle the effects of fresh rainwater mixing from the effects of increased saturation? If not, how significant is the

uncertainty introduced by the different conductivities of rain & pore waters?

A: Disentangling the effects of more resistive rainwater mixing from those of increased saturation is a difficult task. It would require knowing the saturation ratio in the soil and the epikarst prior to the rainfall event. The uncertainty introduced by the different conductivities of rain and pore waters depends upon how high is the difference in resistivity between the rain and the pore water. Figure 12 actually illustrates this issue, based on the Archie's law relationships with successive pore water conductivity values displayed as dashed lines. Theoretically, if the soil has a 0.4 saturation ratio, and a measured resistivity of 1000 Ohm.m, the pore water should be at 30 Ohm.m. An increase in pore water resistivity up to 100 Ohm.m, associated to an increase in saturation up to ∼0.75 is not expected to change the measured resistivity. This is however unlikely to occur at the RCL site given the range of rainwater resistivities as shown in Figure 3d. In any cases, quantifying the amount of uncertainty is out of the scope of this present paper as we demonstrate the inapplicability of converting resistivity into moisture contents at the RCL site, with the available data sets.

R: Section 7.2.2 line 734: Is the fracture drip discharge at PWD1 from a different fracture than the one identified az zone #8 on Fig. 10? It is important to spell this out. Is it assumed that if drip discharge could be monitored from the #8 fracture zone, it would follow a similar pattern to PWD1? The descriptions make it unclear how D1 type regions differ from D3 type regions - they are both fractured regions? Is D3 intermediate between D1 (quick response to precip) and D2 (damped response to precip)? Why does PWD1 correlate with D1 type if it is measuring fracture drip discharge? One would think it should correlated with D3 type? In the text, PWD1 is listed as correlated with D1 regions, but in Fig. 15, PWD1 is shown as correlated with the D3 fractured region. Which is correct? Also, it is possible to estimate rates of recharge for each different type of region? Is it thought that these three types exist in other karst systems as well?

A: This is definitely one point that we will make clearer in the revised version of the

manuscript. The fracture identified in the ERT survey (zone #8) is the fractured zone highlighted in green in Fig. 2c. This is therefore not the one sampled by PWD1, which is parallel to the cutting plane. As for the differences between D1 and D3 type regions and their relationships to PWD1 signal, this should indeed be clarified. D3 region is thought to act as a preferential flow path between superficial layers, such as D1 regions, and deeper layers. Such a fractured zone is likely to be similar to the fracture sampled by PWD1 station. Therefore, quickflows in PWD1 signal reflects mainly variations of water storage in the superficial layers, as it is highlighted by the quick response (<3 hrs) after rainfall. In D3 regions, water exchange between conduits and the porous matrix is likely to explain the seasonal variation. D1 and D2 types of regions correspond to reservoirs that can exist in other karst systems. They can be summarized as an epikarst reservoir and a vadose reservoir, respectively. D3 region images more of a preferential pathway between reservoirs. In any cases, this aspect needs to be explained more clearly in the manuscript. This will be done in this section, as well as in the conclusion, as suggested below by Referee #1.

R: Section 7.2.2 line 739: Clarify the use of the term "subsurface." Is this intended to indicate near-surface regions? It is more commonly used to indicate everything below the surface, so the meaning should be made explicit.

A: You are right, subsurface has multiple meaning. We will change the "surface and subsurface" occurrences into "superficial layers" in the plain text.

R: Section 7.2.2 line 751: This type of general principle for interpreting ERT data should be listed out in one section nearer the beginning of the paper, or at the very start of the analysis section.

A: Right, this will be explained earlier, at the beginning of the discussion.

R: Section 7.2.2 line 778: Again, is the drip discharge at PWD2 & PWD3 coming from the same layers as those identified by ERT analysis as type D2? Or are they following similar behavior patterns but not coming from the same units? This should be explicitly

stated.

A: No, they D2 type regions are not the units sampled by PWD2 and PWD3. However, they reflect similar behavior patterns. We will clarify that in the text.

R: Section 7.2.2 line 791-797: In this paragraph, does "the fractured region" correspond to D3 type? And does "surface layers" correspond to D1 type? Is the primary difference between D1 and D3 that the response to rainfall is more short-lived in D3?

A: Yes, the fractured region corresponds to D3 type, which differentiates from D1 type in showing a damped seasonal behavior, but a high variability in response to rainfall. This is already mentioned in line 736-738.

R: Section 7.2.3 line 875: The preceding section seems to reject both possible causes of the increased resistivity after rainfall. What might then be responsible?

A: As explained line 853-855, we believe that the first proposed possible cause, i.e. the occurrence of artefacts, as those already identified by Descloitres et al. (2008), is more likely to be responsible for the increased resistivity after rainfall. We will add a sentence at the end of Section 7.2.3 to summarize this.

R: Section 7.3 line 886: The previous sentences indicate that the epikarst does not act as a buffer, but this sentence states that the epikarst does act as a buffer. Clarify.

A: The previous sentence does not state that the epikarst has no buffering role, it says that the buffering role of the epikarst is limited. However, ERT results show that the buffering role becomes more important during spring. We will rephrase this section in the manuscript.

R: Figure 15: This is an excellent overview/summary figure. The seasonal variation indicators are particularly helpful. However, in combination with Fig. 2 it is confusing. Fig. 2 seems to indicate that the ERT transect is next to the cave, but Fig. 15 seems to indicate that the cave is directly below the ERT transect. Which is correct? Both figures should be revised to be consistent with each other. The caption for Fig. 15 should also
be much more detailed, with brief summaries of the D1, D2, and D3 dynamics, and the PWD1, 2, and 3 dynamics.

A: Figure 15 is a schematic view of the hydrological processes investigated by the ERT monitoring. The cave is indeed next to the ERT profile, as represented in Fig. 2. This means that, as mentioned above, D3 regions is not sampled by PWD1. This is why we refer to "PWD1 type" flows in Fig. 15. As already said in answers of previous comments, we will make this point clearer in the site description section, and during the discussion of the results, so that no doubts could remain.

R: Section 8: This section is overall clear and concise. The phrasing in this section could be adapted to clarify some of the previous sections.

A: Thanks. We will indeed adapt the phrasing to clarify the message of previous sections.

R: Section 8 lines 939-943: It would be good to think about how the interpretations drawn from ERT data could be further tested hydrologically. Are there other possible interpretations? Could different interpretations be tested against each other? What additional data could support or counter these interpretations?

A: see our answer in the general comment page 1-2.

Technical corrections:

R: There are many small typos, spelling errors, and grammar issues. This paper would benefit from a purely copy-editing oriented revision. A few of the most obvious ones are listed below: l. 17 carbonate l. 39 soil joined with the epikarst l. 54 agree on the dichotomy of matrix and conduit l. 114 the Charlesmont Limestone, which includes l. 115 member acts as l. 117 system ends when l. 189 three groups of joint orientations l. 251 Penman-Monteith l. 256 homogeneously l. 534 inflection l. 740 Klimchouk (2004), defining l. 787 This indicates delayed

A: Thank you for the listing. We will conduct a purely copy-editing oriented revision.

Please also note the supplement to this comment:
https://www.hydrol-earth-syst-sci-discuss.net/hess-2017-477/hess-2017-477-AC1-supplement.pdf

---

## Author Comment (AC2) · 9 Jan 2018

We wish to thank Editor Ty P. A. Ferre for his thoughtful comment that points several improvements to be done in the manuscript. For a better legibility, we have subdivided below the review comment (R) in several paragraphs to which we bring specific answers (A).

R: The authors have addressed a particularly difficult topic within hydrogeology: the dynamics of water flow in complex, layered, heterogeneous regions including relatively large preferential (karstic) pathways. They have applied an accepted and widely used hydrogeophysical method (electrical resistivity tomography) to this task. The strength of the study lies in the unique long-term (three year) data set in a karstic environment.

The challenge, not surprisingly given the complexity of the system, lies in the interpretation of hydrogeophysical data and the transferrability of the method and results. As is fitting for an exploratory application of a new method, the study was conducted in a very well characterized area. It would be worthwhile commenting on how well the ERT data could have been interpreted in the absence of this additional data. This has direct relevance to the use of the ERT method for other, less well-characterized sites. Additionally, the system under study is particularly amenable to study because the water table traverses the known conduits in a typical year, flooding them during winter and running dry in summer. This, again, indicates a wise choice of method for a specific study area. But, it would be worthwhile to comment on this specifically when suggesting that the ERT method could be useful at other karst sites. In other words, it is well known that ERT can only monitor dynamics in as much as it identifies temporal changes in water saturation. How can a reader decide if those conditions are likely to exist at measurable levels at a site before deciding to conduct an ERT survey? A: The unique long-term data set is indeed one of the strengths of this study. One of the messages of the manuscript is however that no strong interpretation could have been drawn without characterizing the site in details, i.e. without the structural and lithological information gathered throughout this study. Although being well characterized, the site might not be ideal to conduct such ERT monitoring. The presence of a pronounced topography and highly dipping geological layers is likely to enhance run-off and infiltration processes that are complex to measure. The ideal site would indeed be a flat karst area with horizontal geological layers. In such cases, the occurrence of groundwater reservoirs within the epikarst would likely be easier to image with ERT monitoring. At the same time, despite the numerous challenges associated with the site under study, we could successfully investigate hydrological processes because of the amount of complementary information gathered throughout the study. In other words, we strongly suggest similar experiment to be conducted in areas where additional information could be collected, i.e. accessible cave systems, implementation of in-situ hydrological measurements, etc. It is also to say that the location of the ERT profile was not set randomly. As mentioned at the beginning of Sect. 5, it required seven preliminary ERT surveys over the study site to select an area with a large spatial distribution in terms of resistivity. This highlights the fact that such ERT monitoring experiment cannot be implemented anywhere. Preliminary studies are necessary to assess the feasibility of such a technique at a given site, and more specifically at the appropriate location within each site. As for measurable levels of temporal changes in water saturation, they are strongly linked with the climatic conditions of each site. Seasonal rainfall averages are a good first indicator to know whether measurable changes in groundwater saturation are likely to occur at a particular site or not. Also, the geological layers being surveyed are important to take into account, which explains why preliminary ERT surveys are important to conduct at any site. We propose a few sentences at the beginning of Sect. 5 to highlight this aspect.

R: The ERT surveys appear to have been very well designed. The combined use of DD and GD surveys is thoughtful and the description of their differences in terms of spatial information and practical limitations is succinctly stated. The use of automated data collection and analysis, incorporating reciprocal measurements where available, gives confidence in the data quality. Similarly, the authors' recognition of temporal variations in contact resistance is noted as this is often overlooked in long term studies. I was impressed by the approach used to correct for temperature effects. However, it isn't clear to me that the heat transport was coupled with water flow. Given the complexity of the hydrologic system, this may not have been possible. But, it would be good to add more detail regarding how the temperature distribution was determined to allow for temperature correction. ( This may be suited to an appendix.) A: We actually removed the description of the temperature correction from a first draft of the paper, as we found that it added unnecessary details and length to the manuscript. But it will indeed be good to add such details as an appendix. As for the model used to calculate the temperature field, it was not possible to couple the heat transport with water flow. Water flows are what we want to identify with the ERT monitoring, which requires not fixing this parameter in the temperature model. In any cases, to our knowledge, there

is no ERT monitoring study that uses a temperature correction that incorporates water flows in the solving of the heat transport. Similarly, the water saturation of each layer is not taken into account in the temperature correction of the resistivity, for the same reasons.

R: It is not clear to me if the resistivity inversion is spatio-temporal or if each time slice was inverted separately. Given that you are looking for changes in time, it seems that spatio-temporal inversion may be more appropriate. But, I would like to have seen some discussion of this choice. It also strikes me that interpreting each time slice independently may be more subject to imposing small scale seasonal variations in areas that are actually not seeing any real variation. It would be very interesting to see if a time lapse inversion results in as good a fit with some areas showing no seasonal EC changes. A: As explained in Sect. 5.2.2 of the present manuscript, we use a time-lapse inversion procedure. This means that a time regularization constraint is used, linking each of the inverted models to the reference model. We will clarify this point in the manuscript to avoid misunderstanding.

R: In the end, I felt that the strongest element of this paper was the structural interpretation. This would be strengthened by more discussion of the process by which the arbitrary conductivity-bounds between regions were determined. It would be far more useful if this were explored automatically, perhaps using clustering techniques to propose alternative structural maps. For me, I think that some discussion of the EC limits is needed for publication. But, the paper would have more impact if this analysis were expanded and potentially seen as the basis for forming competing structural hypotheses. The danger as presented is that the authors may have unconsciously chosen EC limits to confirm their pre-existing structural interpretation. This would, of course, limit the value of all of the work that went into collecting the data. A: It might be important to stress that our main aim was to identify areas of contrasting behavior rather than delineating such areas (i.e. identifying the limits between them). Automatically detect boundaries and even time-lapse changes within ERT images is definitely an

interesting topic. While clustering techniques could certainly bring additional sources of investigation, they also raise several questions about the way they should be implemented. For example, Xu et al. (2017, DOI: 10.1016/j.jappgeo.2017.07.006) has recently addressed clustering problematic within a short time-lapse ERT experiment at the Lascaux cave (France). Their analyses focus on the clustering of an ERT image in several clusters based on resistivity values of a single image. In their study, clustering is however not based on temporal changes within the ERT images. The risk of using clustering based on the resistivity values of a single ERT image is that it neglects the geometries and the dynamics of each cluster. Areas showing similar resistivity would be associated, while they could reflect layers of different lithology, saturation and/or clay ratio. Performing hierarchical agglomerative clustering on the entire time series seems therefore more relevant. Such an approach could for example focus on the clustering of the correlation matrix of all the cells of the resistivity image for all the time steps. Such a method seems interesting but certainly requires further analyses and synthetic modeling to select the best parameters e.g. to pick the best method for calculating the distance between clusters. We believe that this goes further the scope of our paper but we propose to add a few words on these aspects as perspectives in the conclusion. Furthermore, such a technique would not take into account the geometries of the sub-regions within the ERT image. In our case, the highly dipping conductive feature present in the ERT survey is likely to be associated with other, less dipping conductive layers, which does not make sense from a structural point of view. In other words, we believe that our site might not be ideal to start investigating the use of automatic clustering tools, as being too complex in terms of geological structures. This is the main reason why we originally proposed to subdivide our ERT image in 8 sub-regions "based on their average resistivity values and arbitrary thresholds", as explained in the manuscript (Line 557). To clarify this, we propose to add a table with statistical analysis of each cluster (i.e. mean, median, standard deviation through time, etc). We are aware that such an approach is less transportable to other case studies but will contribute to clarify our choice of different clusters

R: Unfortunately, and not unexpectedly given the complexity of the system under study, I found that the hydrologic interpretations were somewhat qualitative. It is interesting to see that there are correlations and delays between responses. But, it doesn't seem to rise to the level of increased understanding of flow dynamics. This may simply be a matter of emphasis - you could highlight what was learned from the ERT that would not have been possible without it. But, it reads more like using your hydrologic insight to give plausible explanations of what you see in the ERT results. Understand, this isn't a strong criticism. I think that this is an advance and shows potential future use of ERT for monitoring dynamics in some karstic systems. But, I think that it is a mistake to make this the emphasis of the paper - starting with the title. Rather, I would focus the paper on the 'hydrostratigraphic' results - showing how you could use the time lapse ERT to identify structure in the subsurface. That could be expanded and extended and then it would be appropriate to say that this interpretation is consistent with what was seen in other hydrologic measurements. As an added benefit, this would allow you to shrink the hydrologic section, which is not as tightly written as the previous sections. A: We appreciate this suggestion. The current version of the manuscript could indeed be reworked to emphasize the 'hydrostratigraphic' aspects. We therefore propose to rearrange the manuscript as suggested, firstly focusing on the ERT results, and then on the structural and hydrological data. This could indeed highlight what could not be learned without the ERT results. However, the interpretation of the ERT dynamics in terms of karst hydrology is for us one of the important aspects of the paper. Especially, the joined analysis between time-lapse ERT results and percolating water measurements is definitely a novel approach that is promising to investigate the sources of distinct in-cave flow types and their lithological/structural constraints. As already explained in our response to Referee #1, we are working on a future paper in which we will focus on a lumped karst modeling of the vadose zone infiltration processes based on the drip discharge data (using the KarstMod modeling platform, from Mazzilli et al., 2017, DOI: 10.1016/j.envsoft.2017.03.015), and their relationships to the ERT data. However, this could not be done without highlighting the role of the lithology and the structures in the

time-lapse resistivity changes. In other words, we see this present manuscript as an original study in which the sources of distinct percolation types are imaged and further linked to geological structures of the karst system under study. We believe that rearranging the section of the manuscript as proposed by Editor Ty P. A. Ferre could clarify our message.

R: In summary, I think that this is a very strong paper and that it should be published in HESS. But, I think that the current emphasis on flow dynamics is not ideal. Rather, it could be a very novel and interesting example of using dynamics to better define structure. This would be most interesting if it could be done automatically, e.g. using clustering tools, and if it led to multiple competing hypotheses that could be further tested in the field. A: Using ERT dynamics within clusters to better define structures would definitely be impressive. But we believe that this goes further the scope of this paper as this would require numerous synthetic modeling for calibrating a working clustering procedure. Moreover, as already pointed out, the site under study might not be ideal to apply and validate such workflows, especially given the complex geological features. Our study however paves the way for such techniques to be tested. Therefore, to strengthen the message of our manuscript, we propose to rearrange the different sections to highlight what was learned from the ERT that would not have been possible without it, as suggested above; the main goal of the paper being to investigate the link between underground structures and percolating water via this long-term ERT monitoring.

Please also note the supplement to this comment:
https://www.hydrol-earth-syst-sci-discuss.net/hess-2017-477/hess-2017-477-AC2-supplement.pdf